# Simulation of transient gusts on the NREL5 MW wind turbine using the U-RANS-solver THETA

Annika Länger-Möller[1]

[1]DLR e.V.; Lilienthalplatz 7; 38108 Braunschweig

**Correspondence:** Annika Länger (annika.laenger@dlr.de)

**Abstract.** A procedure to propagate longitudinal transient gusts through a flow field by using the resolved-gust approach is implemented in the U-RANS solver THETA. Both, the gust strike of a $1 - \cos()$-gust and an extreme operating gust following the IEC 61400-1 standard are investigated on the generic NREL 5 MW wind turbine at rated operating conditions. The impact of both gusts on pressure distributions, rotor thrust, rotor torque, and flow states on the blade are examined and quantified. The
5   flow states on the rotor blade before the gust strike, at maximum and minimum gust velocity are compared. An increased blade loading is detectable in the pressure coefficients and integrated blade loads. The friction force coefficients indicate the dynamic separation and re-attachment of the flow during the gust. Moreover, a verification of the method is performed by comparing the rotor torque during the extreme operating gust to results of FAST rotor code.

## 1 Introduction

The origins of applying Computational Fluid Dynamics (CFD) to wind turbine rotors date back to the $1990^{th}$ when Soerensen and Hansen (1998) applied EllipSys3D to a wind turbine. Soerensen and Hansen (1998) solved the Reynolds Averaged Navier-
15   Stokes (RANS) equations and applied the Menter SST $k-\omega$-turbulence model to a full scale wind turbine. In 2002 the National Renewable Energy Laboratory (NREL) performed an Unsteady Aerodynamic Experiment (UAE) (Hand et al., 2001) which has long been the reference for several CFD computations. For example, Johansen et al. (2002) presented a Detached Eddy Simulation (DES) on the NREL UAE phase VI blade to demonstrate the capabilities of predicting flow separation. Soerensen and Schreck (2012) developed a delayed DES to investigate whether the simulation approach could be improved. Furthermore,
20   the experiment has been widely used for U-RANS solver validation for example by Duque et al. (2003); Le Pape and Lecanu (2004); Yelmule and Anjuri (2013); Lynch and Smith (2013); Oe et al. (2014); Länger-Möller (2017).
Jonkman et al. (2009) developed the generic NREL 5 MW wind turbine. Through its open access documentation, the NREL 5 MW wind turbine is established as reference and validation test case for single- and multi physic test cases. For example,

Chow and van Dam (2012) focused on the prediction of aerodynamic features of the wind turbine. Furthermore, they investigated the impact of fences on the flow separation in the inboard region. Full aero-elastic computations were performed by Bazilevs et al. (2011) for an isolated rotor and Hsu and Bazilevs (2012) respecting also the tower and nacelle. Bazilevs et al. (2011) and Hsu and Bazilevs (2012) modelled the aerodynamics with an U-RANS method and the structure with shell elements. The structure properties represented the material properties of the blade. The resulting aerodynamic characteristics and blade tip deflections were good when compared to the NREL 5 MW documentation and FAST.

In the past years, the growing computer power enabled the geometry resolved simulation of wind turbines including the sites with Computational Fluid Dynamics (CFD). Studies have been performed for example by Schulz et al. (2016) or Murali and Rajagopalan (2017) who used an Unsteady Reynolds-Averaged Navier Stokes (U-RANS) solver to perform according studies. Moreover, hybrid Large Eddy Simulation(LES)-RANS approaches are implemented to analyse the behaviour of wind turbines in a complex terrain as for example presented by Castellani et al. (2017) who also considered the unsteady atmospheric inflow conditions.

The challenges of correctly predicting uncertainty of the fluctuating wind loads is a research field on its own. For example, Bierbooms and Drag (1999) or Suomi et al. (2013) investigated wind fields to better understand the shape of wind gusts. Matthäus et al. (2017) argued that a detailed understanding of wind fields is not necessary. Matthäus et al. rather respected unknowns of all parts of the wind turbine life cycle as for example changes in the blade shape due to production tolerances, ageing, or the wind field and summarized them in uncertainty parameters to estimate the effective power outcome and rotor loads. Mücke et al. (2011) proved that turbulent wind fields are not distributed Gaussian as assumed in the International Electrotechnical Commission Standard (IEC 61400-1). A similar conclusion was drawn by Graf et al. (2017) who investigated whether the 50 year loads as defined in the IEC 61400-1 adequately fulfil their purpose by applying different approaches of probability prediction to the generic NREL 5 MW turbine using the FAST rotor code.

The aerodynamic interferences between the unsteady wind conditions and wind turbines are of major importance for the prediction of fatigue loads and the annual power production. Therefore, it is part of the certification computation for each wind turbine. Nevertheless, the detailed investigation of isolated effects as the 50 year Extreme Operating Gust (EOG) on the flow of a wind turbine using high fidelity methods as CFD is rare even though the blade loads resulting from the extreme load cases are dimensioning load cases. In the case of vertical axis wind turbines Scheurich and Brown (2013) analysed the power loss of a wind turbine subjected to a sinusoidal fluctuation in wind speed. However, compared to the EOG amplitudes were small. Horizontal axis wind turbines which are hit by an EOG as defined in the IEC 61400-1 were presented by Sezer-Uzol and Uzol (2013). The wind turbine under consideration was the NREL phase VI rotor with a wind speed of 7 m/s using the panel code AeroSIM+. The impact of the gust was then evaluated in terms of rotor thrust, torque and wake development. Preceding this study, Bierbooms (2005) examined the flap moment of wind turbine blades which were subjected to a gust with extreme raise, using the wind turbine design tool Bladed. Bladed is an aero-elastic software by Garrad Hassan for the industrial design and certification of wind turbines (DNVGL, 2017). Other examples of aero-elastic simulation tools for wind turbine design are HAWC2 (Larsen and Hansen, 2015) or FAST (Jonkman, 2013) which all include at least a blade element momentum (BEM) method to represent the aerodynamics, a multi-body dynamics formulation to represent the structure, and an algorithm for

rotational speed control. All three of them provide a possibility to compute EOG cases fully-multidisciplinary on the basis of linearized aerodynamic and structure models.

Even though the literature on gust simulations on wind turbines is rare, some research has been conducted in the field of aerospace science. Kelleners and Heinrich (2015) and Reimer et al. (2015) presented two approaches which are implemented in the U-RANS solver TAU (Schwamborn et al., 2006) to apply vertical gusts on airplanes: the velocity-disturbance approach and the resolved-gust approach. The velocity-disturbance approach adds the gust velocity to the surface of the investigated geometry. It enables the analysis of the resulting forces on the geometry surface but prevents the feedback of the structure response on the flow field and the gust shape. The resolved-gust approach overcomes the disadvantages of the one-way interaction in the velocity disturbance approach by propagating the gust through the flow field with the speed of sound. But it ignores that the gust transport-velocity usually differs from the speed of sound. The validity of both implementations was demonstrated by the time history of the position of the centre of gravity, pitch angles and load factors necessary for keeping the flight path of an aircraft constant.

In the so-called field approach Parameswaran and Baeder (1997) added the gust velocity to the grid velocity of the computational grid to all cells with

$$x \leq u \cdot t \tag{1}$$

wherein $x$ is the coordinate in flow direction, $u$ the gust transport velocity and $t$ the physical time. This approach allows the definition of a gust transport-velocity and the analysis of the two-way interaction between gust, structure, and wake. Nevertheless, it requires a severe manipulation of the velocity field regardless of the flow solution that is produced by the wind turbine.

The simulation of unsteady inflow conditions of wind turbines in CFD implies several challenges: The simulation of a wind turbine including the tower is, itself, an in-stationary problem which needs the computation of several rotations to obtain a periodic solution. Superposed by sheared inflow profiles and in-stationary (stochastic) inflow conditions, periodicity can never be gained because the same flow state never occurs twice. Moreover, a computation in which the rotor motion is adapted to the actual rotor forces using a strong coupling approach as proposed by Sobotta (2015) should be included in the computation. By using strong coupling between the U-RANS solver Fluent and a pitch control algorithm for the rotor-motion Sobotta has been able to implement a simulation-procedure of turbine start-up. Heinz et al. (2016) performed the computation of an emergency shut-down of a turbine by using the incompressible U-RANS solver EllypSys3D and by neglecting the tower throughout the aerodynamic computations. Additionally, Heinz et al. considered the rotor mass and inertia by coupling the U-RANS solver with the aeroelastic code HAWC2.

The validation of the resolved-gust approach in the DLR U-RANS solver THETA (Löwe et al., 2015; Länger-Möller, 2017) is presented herein. To reduce the complexity of the problem and emphasize the quality of the resolved gust approach, the NREL 5 MW wind turbine is chosen to operate in shear-free conditions. Moreover, the possible interferences with the structure response and speed controllers are reduced by using infinite rotor mass and inertia. Speed control algorithms are also neglected. As gust, a $1 - \cos()$ shaped gust which lasts about 7 s and the EOG following the IEC 61400-1 standard are chosen. As results

rotor thrust and rotor torque, pressure distributions, friction force coefficients and the wake-vortex transport are evaluated. The rotor torque during the EOG is validated against FAST.

## 2 Numerical methods

### 2.1 Flow solver THETA

DLR's flow solver THETA is a finite volume method which solves the incompressible Navier-Stokes (NS) equation on unstructured grids. The grids can contain a mix of tetrahedrons, prisms, pyramids and hexagons. The transport equations are formulated on dual cells, which are constructed around each point of the primary grid. Therefore, the method is cell-centred with respect to the dual grid. The transport equations are solved sequentially and implicitly. The Poisson equation which links velocity and pressure is either solved by the SIMPLE algorithm for stationary problems or the projection method for unsteady

simulations. With the projection method the momentum equations are first solved with an approximated pressure field. The pressure field then is corrected with a Poisson equation to fulfil continuity. Pressure stabilization is used to avoid spurious oscillations caused by the collocated variable arrangement.

The technique of overlapping grids (Chimera) is used to couple fixed and moving grid blocks. The method was developed by Pan and Damodaran (2002) for structured grids or Zhang et al. (2008) for unstructured grids for the application in incompress-

ible flow problems. It has been implemented to THETA by Kessler and Löwe (2014). The interpolation between the different blocks at interior boundaries is integrated in the system of linear equations on all grid levels of the multi-grid solver leading to an implicit formulation across the blocks. This procedure was identified to be crucial for achieving fast convergence of the Poisson equation.

Implicit time-discretization schemes of first order (implicit Euler) or second order (Crank Nicolson, BDF) are implemented.

The temporal schemes are global time stepping schemes. A variety of schemes from first order upwind up to second order linear or quadratic upwind or a central scheme and the low dissipation, low dispersion scheme (Löwe et al., 2015) are implemented. Throughout this study, the second order central scheme is used.

The THETA code provides a user interface for setting complex initial and boundary conditions using the related C functions. This guarantees a high flexibility on the definition of boundary conditions and a straightforward modelling of very specific test

cases. For example, the functions enable the prescription of gusts at the inflow boundary condition which are then propagated through the flow field. Moreover, all physical models are separated from the basis code. Therefore, new physical models can be implemented without modification of the base code.

For turbulence modelling the commonly used Spalart-Allmaras, $k - \epsilon$, $k - \omega$ or Menter-SST models are available. Since the early U-RANS computations of wind turbines the Menter-SST turbulence model is used (Soerensen and Hansen, 1998) for

wind turbine applications. Recently, Länger-Möller (2017) confirmed this finding during the THETA validation by comparing the results of common one- and two equation turbulence models to the NREL UAE phase VI experiment. Hence, the Menter-SST turbulence model is applied throughout the present study. Moreover, according to the studies in Länger-Möller (2017) a time step of $\delta t = 0.006887052\,\mathrm{s}$ which is equivalent to a rotor advance of $\Psi = 0.5°$ per time step is chosen. As time stepping

scheme, the Eulerian implicit scheme for the temporal discretization is chosen. To ensure convergence in every time step, a residual of less than $10^{-5}$ has to be reached. Moreover, the solver has to perform at least 20 iterations per time step in all equations. Due to efficiency reasons, the maximum number of iterations per time step has been limited to 100.

## 2.2 FAST

The comprehensive rotor code FAST (Jonkman, 2013) is a modular software framework for computer aided engineering (CAE) of wind turbines. FAST provides a coupling procedure to compute time-dependent multi-physics relevant for wind turbine design. By the means of different modules, FAST is able to respect different physical models and turbine components in the computations. The aerodynamics are represented by a blade element momentum method (BEM) which is based on profile polars for drag, lift and momentum.

In the present case, most parameters remained on the default of NREL's v8.16.00a version for both FAST and the NREL 5 MW wind turbine. Few parameters had to be adjusted. As there are: the variable-speed control has been turned off to ensure a constant rotational speed as in the U-RANS computation. The blade stiffness has been increased to the order of $10^{29}$ Nm$^2$ per blade element to obtain a stiff blade. To ensure a shear-free inflow profile, the constant wind profile type without dynamic inflow model was selected. In FAST, the EOG started after the computation of 8 s. The analytical inflow profile was included

as $x$-velocity in the IECWind file. The other wind directions were equal to 0.

## 3 Geometry

### 3.1 NREL 5 MW wind turbine

The NREL 5 MW turbine (Jonkman et al., 2009) is a three bladed wind turbine with a rotor radius of 63.0 m and a hub height of 90 m. The rotor has a cut-in wind speed of $v_{ci} = 3$ m/s and a rated wind speed of $v_{rated} = 11.4$ m/s. Cut-in and rated

rotational speeds are $\omega_{ci} = 41.4\,°/s$ and $\omega_{rated} = 72.6\,°/s$, respectively. The blades are pre-coned and the rotor plane is tilted about $\beta = 5.0\,°$ and not yawed. Along the non-linearly twisted blade 7 different open-access profiles are used.

Due to the narrow gap between rotor and nacelle a valid chimera overlap-region could not be achieved in that region. Thus the nacelle of the NREL 5 MW turbine is neglected while the tower is respected. This approach leads to an error in the flow prediction behind the rotor hub but is supposed to have no impact on the blade loads.

The gust simulation is based on the rated wind speed $v_{rated} = 11.4$ m/s. Air density of $\rho = 1.225$ kg/m$^3$ and the kinematic viscosity of $\nu = 1.82 \cdot 10^{-5}$ m$^2$/s is used. To isolate the gust impact of the rotor loads, a shear-free velocity profile is considered throughout the computation.

### 3.2 Grid characteristics

The computational grid consists of 3 parts. The first part contains the three rotor blades, stubs and the rotor hub. On the

blade surface, a structured grid with $156 \times 189$ elements in span-wise and chord-wise direction was generated. The boundary

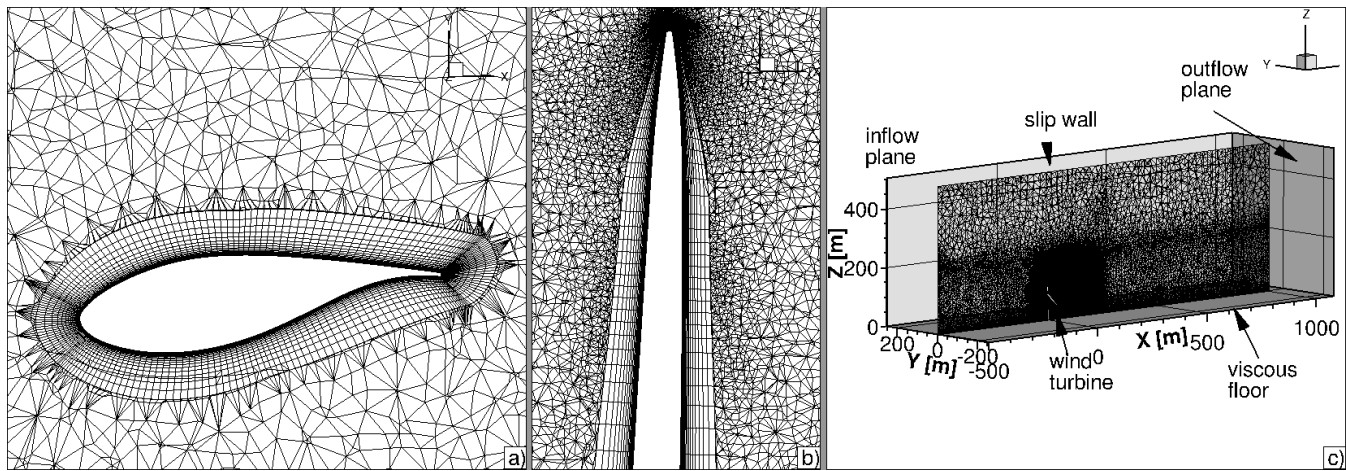

**Figure 1.** Computational grid setup; a) Chord wise distribution; b) span wise distribution in blade tip region; c) Cut through the flow field and boundary conditions

layer mesh of the blades consists of 49 hexagon layers in an O-O-topology. The height of the wall-next cell is $\delta = 3 \cdot 10^{-6}\,\mathrm{m}$ along the entire blade, ensuring $y^+ \leq 1$. Figure 1a) and b) give an impression of the chord-wise and span-wise grid resolution, respectively.

The second part of the grid has the shape of a disk and contains the entire rotor. The disk measures $D = 166.7\,\mathrm{m}$ in diameter
and has a depth of $26.7\,\mathrm{m}$. It is filled with tetrahedrons with an edge length between $0.002\,\mathrm{m}$ and $0.9\,\mathrm{m}$. The entire disk is used as chimera child grid for the overlapping grid technique and contains approximately $11.63 \cdot 10^6$ points.

The chimera parent grid has the dimensions of $504 \times 504 \times 1512\,\mathrm{m}^3$ in width, height, and length. It contains a boundary layer grid of the floor, the tower, and a refined grid region to resolve the rotor wake up to $3R$ downstream.

The 54 prism layers, used to resolve the boundary layer of the viscous floor, have a total height of $H = 5\,\mathrm{m}$ with a wall-next cell
height of $\delta = 3 \cdot 10^{-5}\,\mathrm{m}$. This meshing strategy enables the comparison of future gust computations that include analytically defined velocity-profiles of neutral atmospheric boundary layer flows. The tower surface-grid is meshed structured in a height below $5\,\mathrm{m}$ with 54 point in height and 180 points in radial direction. Above $5\,\mathrm{m}$, a triangulated unstructured grid is generated with the maximum edge length of $0.55\,\mathrm{m}$. As the tower surface is modelled as slip-wall, tetrahedrons are built directly on the tower surface.

In the chimera parent grid, the edge length of the cells continuously grow from very small in the rotor-tower and wake region to rather large close to the farfield boundaries. The entire chimera parent grid contains approximately $13.25 \cdot 10^6$ points.

In figure 1c) the entire chimera setup is displayed and the boundary conditions are indicated. The upwind and downwind boundaries are defined as inflow and outflow respectively. At the inflow boundary surface, the turbulence quantities and inflow velocities are prescribed. Later, also the gust profile is introduced at this boundary. The floor is defined as viscous wall. The
surfaces on top, left and right of the flow domain are defined as slip wall.

## 4 Gust modelling

### 4.1 The resolved-gust approach

The procedure of applying the gust to the flow field starts by computing the flow field around the wind turbine until the flow field and the global rotor loads have become periodic. For the NREL 5 MW turbine in the given setup 9 revolutions are required. Then, the inflow velocity on the inflow boundary is modified according to the velocity change described in section 4.2 or 4.3. The computation is continued so that the gust is propagated through the flow field. In the approach by Kelleners and Heinrich (2015) and Reimer et al. (2015) using TAU (Schwamborn et al., 2006) to solve the compressible RANS equations, the gust is transported with the speed of sound. In their approach, as well as in the present paper, the computation has been run at least until the gust has entirely passed the geometry in question but can be continued as long as wished by the user. The restrictions to ensure a loss-free transport of the gust velocity on the resolved-gust approach named by Kelleners and Heinrich (2015) or Reimer et al. (2015) are

- a fine grid upstream of the geometry in question

- a fine time step.

As THETA is an incompressible solver, the speed of sound is infinite. In addition to the strong implicit formulation and the choice of boundary conditions that prevent the flow from escaping sideways, this leads to a spread of the gust velocity through the flow field instantaneously. If the same gust velocity is added to the constant inflow condition in every point in the inflow plane and the boundary conditions are chosen as specified in section 3.2, the transport of the gust velocity will be loss-free and instantaneous through the entire domain and on the farfield boundaries. Hence, the restrictions by Kelleners and Heinrich (2015) and Reimer et al. (2015) concerning the grid resolutions are obsolete while a fine time step is required to ensure numerical stability.

To analyse the resolved gust approach in the incompressible U-RANS solver THETA, the inflow velocity profile is shear-free and the gust velocity remains independent of the height above ground.

### 4.2 Cosines gust

The $1 - \cos()$ gust is modelled in analogy to the EASA certification standard (EASA, 2010) as

$$u(t) = u_g \left( 1 - \cos \left( \frac{8\pi}{H} \right) \right) \tag{2}$$

with $u(t)$ and $u_g$ as the time dependent velocity and the gust velocity, respectively and $H$ as gust gradient. In the work presented, $H$ is chosen to generate a non-compressed sinusoidal gust

$$H = \frac{8\pi}{t - T_S} \tag{3}$$

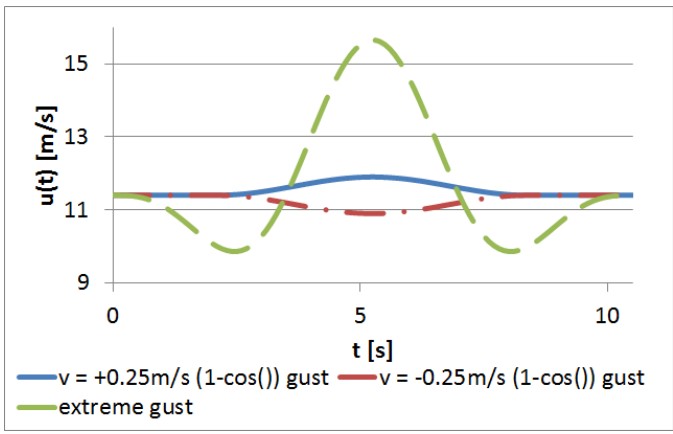

**Figure 2.** Inflow velocity in dependence of physical time $t$

wherein $t$ represents the actual physical time and $T_S$ is the time at which the gust starts. Inserting equation 3 in equation 2 the following definition of the gust results:

$$u(t) = \begin{cases} u_g(1 - \cos(t - T_S)) & \text{if } T_S \leq t < T_S + T_g \\ u & \text{if } t \leq T_S \text{ or } t \geq T_S + T_g \end{cases} \tag{4}$$

Wherein $T_g$ is the duration time of the gust. The gust velocity $u_g$ is defined as $+0.25\,\text{m/s}$ representing a gust and $-0.25\,\text{m/s}$
representing a sudden calm. In both cases, the maximum change in wind speed is $0.5\,\text{m/s}$ or $4.4\,\%$ at rated wind speed of the NREL 5 MW turbine. The turbulence intensity of the $1 - \cos()$-gust is $2.5\,\%$ which is in the order of the atmospheric turbulence intensity that Schaffarczyk et al. (2017) found in a field measurement campaign on a horizontal axis wind turbine. The resulting gust profile of the present study is displayed in figure 2.

### 4.3 Extreme operating gust

The time dependent velocity change of the EOG is modelled following the IEC 61400-1 standard.

$$u(z,t) = u(z) - 0.37 \cdot u_g \sin\left(\frac{3\pi \cdot t}{T_g}\right) \cdot \left(1 - \cos\left(\frac{2\pi \cdot t}{T_g}\right)\right). \tag{5}$$

Wherein $T_g = 10.5\,\text{s}$ is the characteristic time as defined in the IEC 61400-1 (2005), $t$ the physical time simulated, $u(z)$ the velocity profile depending on the height and $u_g$ the gust velocity. The latter is defined as

$$u_g = 3.3 \left(\frac{\sigma_1}{1 + 0.1 \left(\frac{D}{\lambda_1}\right)}\right) \tag{6}$$

and is $5.74\,\text{m/s}$ in the given case. In equation 6, $\sigma_1 = 0.11 \cdot u_{hub}$ is the standard turbulence deviation, $\lambda_1 = 42\,\text{m}$ the turbulence scale parameter and $D$ the rotor diameter. $u_{hub}$ represents the velocity at hub height. The velocity $u_{e1}$ is the average over 10

minutes with a recurrence period of 1 year. It is defined as

$$u_{e1} = 1.12 \cdot u_{ref} \left( \frac{z}{z_{hub}} \right)^{0.11} \tag{7}$$

with the reference velocity $v_{ref} = 50\,\mathrm{m/s}$. It is defined in the IEC 61400-1 standard for a wind turbine of the wind class A1. In a shear-free flow, the inflow velocity is constant with height and thus $u(z)$ reduces to $u$ and $u(z,t)$ to $u(t)$. By additionally

entering equation 7 in equation 5 one obtains the final gust definition

$$u(t) = u - 0.37 \cdot u_g \sin\left( \frac{3\pi \cdot t}{T_g} \right) \cdot \left( 1 - \cos\left( \frac{2\pi \cdot t}{T_g} \right) \right). \tag{8}$$

The resulting gust profiles of the EOG in comparison to the moderate $1 - \cos()$-gust is visualized in figure 2.

## 5 Results

### 5.1 Constant inflow conditions

As described in section 4.1 a periodic flow field with periodic rotor loads is mandatory as starting conditions for computing gusts that act on wind turbines. The resulting time history of rotor thrust $F_x$ and rotor torque $M_x$ for constant inflow conditions over revolution 6 to 10 are displayed in figures 3 and 4 with the red line. In both figures the periodic behaviour of a periodic flow field is visible as well as the typical $3/rev$-characteristic of a rotor-tower configuration of the wind turbine. By averaging rotor thrust $F_x$ and torque $M_x$ about 4 revolutions, one obtains $738.9\,\mathrm{N}$ and $4.15\,\mathrm{MNm}$, respectively. Compared to the reference of

Jonkman et al. (2009) at rated conditions the values deviate about approximately $-3.77\,\%$ and $-0.98\,\%$, respectively. Imiela et al. (2015) achieved a rotor thrust of $786\,\mathrm{kN}$ and a torque of $4.4\,\mathrm{MNm}$ in their studies with the compressible U-RANS solver TAU for the stiff-bladed NREL 5 MW turbine.

The agreement between the U-RANS computations performed with THETA, TAU and the reference documentation of the NREL 5 MW wind turbine (Jonkman et al., 2009) is excellent. Thus, the numerical setup is validated successfully.

Between $\Psi = 3060°$ and $\Psi = 3240°$, in the $8^{th}$ revolution, high-frequency oscillations occur in the THETA computation. In the specific time step, the Poisson equation for pressure correction has not converged in the maximum number of iterations. Nevertheless, the interference subsides in the following rotor rotations and is sufficiently small. Thus, the reason for this oscillations is of minor importance in the context of this paper.

If the wind turbine operates in uniform flow conditions, a $3/rev$-characteristic is found in both $F_x$ and $M_x$, which is caused

by the tower blockage effect. Moreover, the constant amplitudes around a steady mean value of both, $F_x$ and $M_x$ indicate that the flow field has converged. The converged state includes the boundary layer that developed on the viscous floor of the flow domain. Over the length of the entire flow domain, the boundary layer achieved a thickness of approximately 1m at the end of the flow domain. This is far below the rotor area and does not affect the rotor characteristics or wake development during the computation. Hence, the gust as defined in equation 4 or 8 can be applied in the next step.

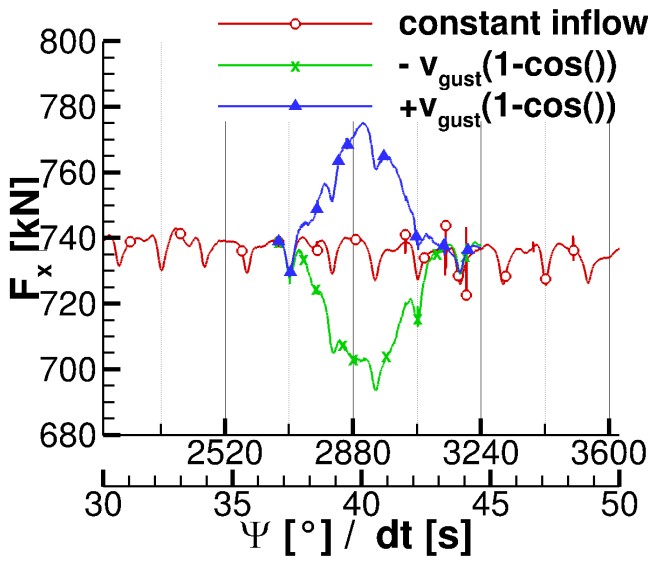

**Figure 3.** Rotor thrust $F_x$ during the gust

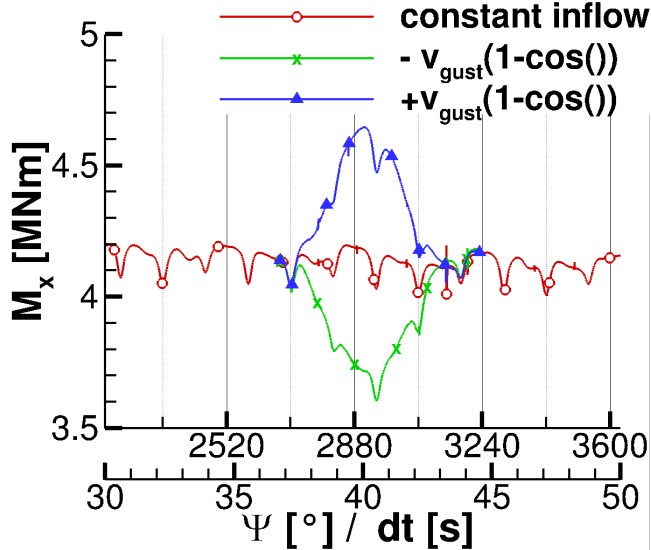

**Figure 4.** Rotor torque $M_x$ during the gust

### 5.2 Cosines gust

The impact of the $1 - \cos()$- gust on rotor thrust $F_x$ and rotor torque $M_x$ during the gust is evaluated by comparison to uniform inflow conditions. $F_x$ and $M_x$ are displayed in figure 3 and 4, respectively. Therein, the period between $30\,\text{s}$ and $50\,\text{s}$ is displayed while the gust operates between $T_S = 37\,\text{s}$ and $T_S + T_g = 44\,\text{s}$. Thus, it lasts approximately 1.5 rotor revolutions.

As expected, the gust velocity spreads over the entire field immediately and also affects rotor thrust and rotor torque instantaneously. In the case of gust and calm no hysteresis effect is found as rotor thrust and rotor torque recover immediately after the gust. This is visible in both figures 3 and 4 as the curve of constant inflow conditions is matched right after $44\,\text{s}$. The symmetric response of rotor thrust and rotor torque to the gust is caused by the modelling assumptions of a stiff blade and a missing speed control algorithm.

During the gust, $F_x$ and $M_x$ follow the modification of the inflow condition. Hence, for a positive gust velocity $u_g$ (equation 4) rotor loads increase in a $1 - \cos()$-shape while they decrease in the same manner for a negative gust velocity. Additionally, the tower blockage effect is superposed on the blade loads and remains detectable in the blade load development. In the case of $u_g = +0.25\,\text{m/s}$, the tower blockage effect reduces the time that the rotor experiences maximum loads as is visible at approximately $t = 41\,\text{s}$. A sharp drop in both, rotor thrust and rotor torque, is visible. This drop is due to the tower blockage effect and

would have appeared at a different instance of the gust if the rotor position at the gust starting time was different. Nevertheless, during the calm with $u_g = -0.25\,\text{m/s}$ the tower blockage leads to an additional decrease in rotor thrust and rotor torque at $t = 41\,\text{s}$. The rapid changes in rotor thrust and rotor torque indicate the fast load changes on the blade which increase fatigue loads.

**Table 1.** Gust induced peak loads on the rotor during the $1 - \cos()$-gust in relation to the constant blade load

|  | $\delta F_x$ [%] | $\delta M_x$ [%] |
|---|---|---|
| $u_g = +0.25\,\mathrm{m/s}$ | +5.6 | +12.9 |
| $u_g = -0.25\,\mathrm{m/s}$ | -5.6 | -12.4 |

**Table 2.** Averaged rotor loads during the $1 - \cos()$-gust in relation to the constant blade load

|  | $\delta \overline{F_x}$ [%] | $\delta \overline{M_x}$ [%] |
|---|---|---|
| $u_g = +0.25\,\mathrm{m/s}$ | +2.4 | +5.7 |
| $u_g = -0.25\,\mathrm{m/s}$ | -2.1 | -4.6 |

Table 1 lists the relative differences in $F_x$ and $M_x$ during the gust, computed by equation 9. In equation 9 the subscript $max$ indicates the extreme rotor loads and the over-line the averaged rotor loads under constant inflow conditions of section 5.1

$$
\begin{aligned}
\delta F_x &= 100 \cdot \left( \frac{F_{max}}{F_x} - 1 \right) \\
\delta M_x &= 100 \cdot \left( \frac{M_{max}}{M_x} - 1 \right).
\end{aligned}
\tag{9}
$$

Additionally, the relative difference in the averaged blade load is computed by first integrating rotor thrust and rotor torque during the gust excitation and then computing equation 9. The result of the gust peak load is listed in table 1 while the integrated loads are contained by table 2.

In both tables it can be seen that a reduction of the wind speed due to a calm or the increase of the wind speed with the same amplitude leads to very similar absolute changes in rotor thrust and rotor torque. By comparing the values of table 1 and 2 it is

also found that the absolute peak loads are 2.3 times larger than the averaged loads. Thus, the use of maximum loads during a $10\,\mathrm{min}$-interval is inevitable for a computation of equivalent fatigue loads while averaging the loads is not appropriate.

It is also important to note that the rotor loads return to the values of constant inflow conditions right after the gust ended. This indicates that there are neither reflections nor numerical oscillations in the flow field which lower the numerical accuracy. In summary, the behaviour of rotor thrust $F_x$ and rotor torque $M_x$ is as expected. The increased wind speed causes higher thrust

and momentum and vice versa while the amplitude is identical for the increase and decrease in wind speed.

By considering blade deflections and changes to the rotational speed in future aero-elastic computations, the resulting rotor torque and rotor thrust will change.

### 5.3 Extreme operating gust

Figure 5 presents rotor thrust $F_x$ and rotor torque $M_x$ during the EOG excitation in comparison to the constant inflow condi-

tions. Moreover, the rotor torque that is computed by FAST for a stiff blade and constant rotational speed is displayed. The EOG lasts about $10.5\,\mathrm{s}$ or 2 rotor revolutions. In comparison to the rotor loading during the $1 - \cos()$-gust (section 5.2) the

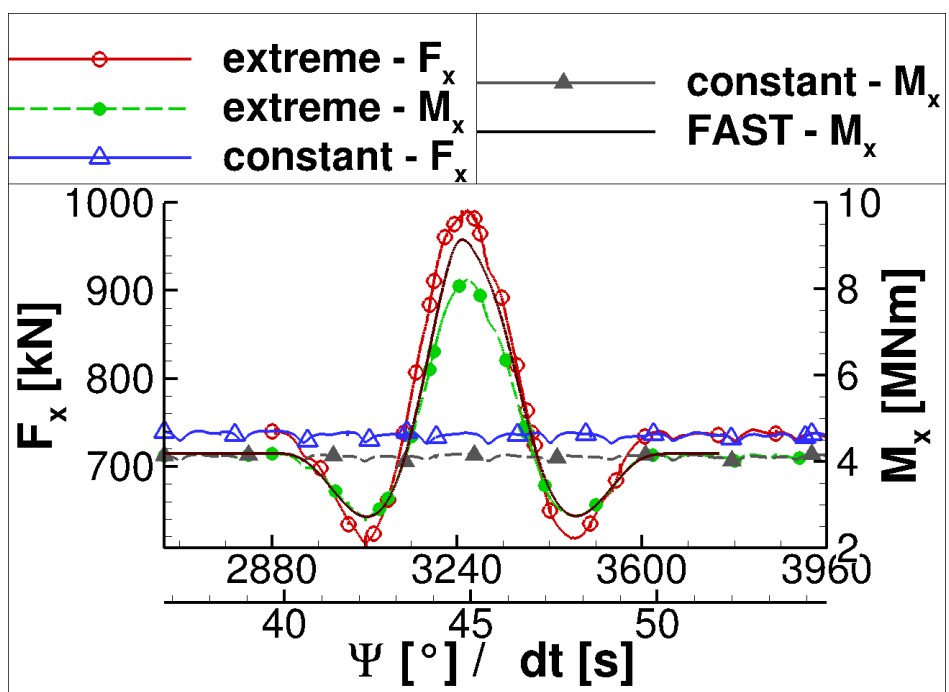

**Figure 5.** Rotor thrust $F_x$ and torque $M_x$ during the gust

tower blockage effect becomes negligible. Hence, the starting position of the rotor is less important for the computation of maximum loads. This is also seen in the FAST result. By comparing the rotor torque during the gust of THETA to the one of FAST it is found that both values coincide exactly before $t = 43.5\,\mathrm{s}$ and after $t = 46.5\,\mathrm{s}$. Between this two time stamps, FAST predicts significantly higher loads than THETA. Moreover, the EOG load at the maximum gust velocity is increased in comparison to THETA. The differences result from the flow characteristics of the blade. THETA predicts large areas of flow separation as response to accelerated velocity after $t = 43.5\,\mathrm{s}$. The flow reattaches over most of the blade after $t = 46.5\,\mathrm{s}$ when the velocity slowed down sufficiently. It is most likely that the profile polars that the BEM of FAST relies on is not able to reproduce the instationary flow behaviour of the given case.

The maximum velocity $u_{max}$ during the gust is about $15.65\,\mathrm{m/s}$ and the minimum velocity $u_{min}$ is $9.94\,\mathrm{m/s}$ which are $137\,\%$ and $87\,\%$ of the values at rated wind speed. The changes in rotor thrust and rotor torque, computed by equation 9, are given in table 3. It is shown that the rotor torque is decreased by $36\,\%$ during the calm that precedes or follows the velocity maximum and increased about $100\,\%$ during the gust peak. The changes in rotor thrust are smaller even though the amplitudes of load change are significant as well.

To analyse the flow state on the blade during the gust two instances have been chosen: after $t_{min} = 2.5s + T_S = 42\,\mathrm{s}$ (minimum gust velocity) and $t_{max} = 5.6s + T_S = 45\,\mathrm{s}$ (maximum gust velocity). Figure 6a) and b), respectively, display the rotor positions in the instances investigated. In both figures blade number 1 is coloured in black. At $t_{min}$, when the gust is at its minimum velocity, blade number 1 is right in front of the tower and additionally experiences the tower blockage effect. Conversely, at

**Table 3.** Gust induced peak load during the EOG on rotor in relation to the constant blade load

|  | $\delta u \, [\%]$ | $\delta F_x \, [\%]$ | $\delta M_x \, [\%]$ |
|---|---|---|---|
| $u_g(min)$ | $-13$ | $-17.3$ | $-36.0$ |
| $u_g(max)$ | $+37$ | $+35.0$ | $+100.2$ |

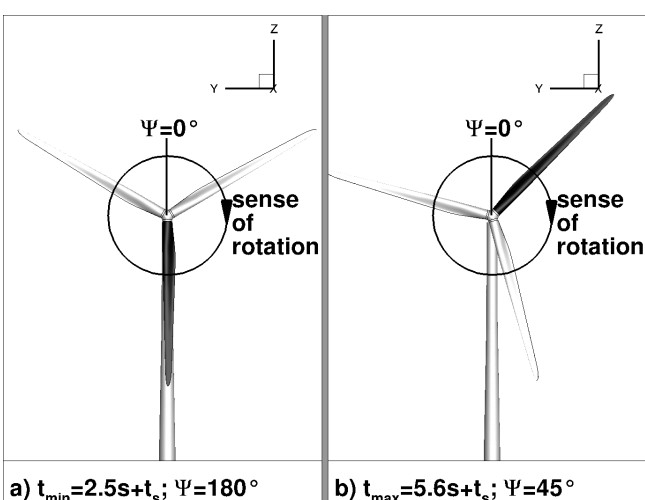

**Figure 6.** Rotor position at minimum (left) and maximum (right) gust velocity; Black blade is blade number 1

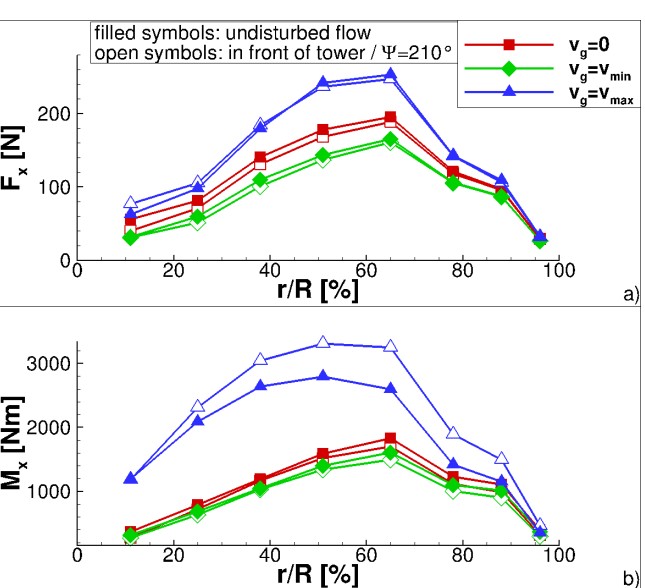

**Figure 7.** Span-wise distribution of a) rotor thrust and b) rotor torque for different azimuth positions at constant inflow and during the gust

$t_{max}$, when the gust is at its maximum velocity, blade number 1 is in free stream conditions while the flow on blade number 3 enters the tower blockage region. The impact of the tower blockage during the constant inflow conditions at $t_{min}$ and $t_{max}$ is visible in the radial distribution of rotor thrust and rotor torque (figure 7).

In accordance to figure 5 the overall rotor loading is reduced when the blade experiences minimum gust velocity. Conversely,
5   a significant increase in rotor loading is observed when the blade experiences the maximum gust velocity. At all times, the rotor thrust is reduced only slightly by the tower blockage effect. Moreover, only the inboard part of the rotor appears to be affected by the tower blockage. Contrariwise, the rotor torque is affected by the tower blockage in the outer part of the rotor. Besides, the tower blockage effect generally has only small impact on the rotor torque at wind velocities smaller than rated wind speed. With the maximum gust velocity, the blade at $\Psi = 210°$ experiences a strong tower blockage effect that reduces
10   the rotor torque up to $29\%$ (radial section $r/R = 65\%$). The reason is the different separation behaviour at the trailing edge of the blade which are investigated through pressure coefficient distributions and friction force coefficients at three radial sections: an inboard section at $r/R = 10\%$, a mid-section $r/R = 50\%$ and an outboard section at $r/R = 90\%$. They are displayed in

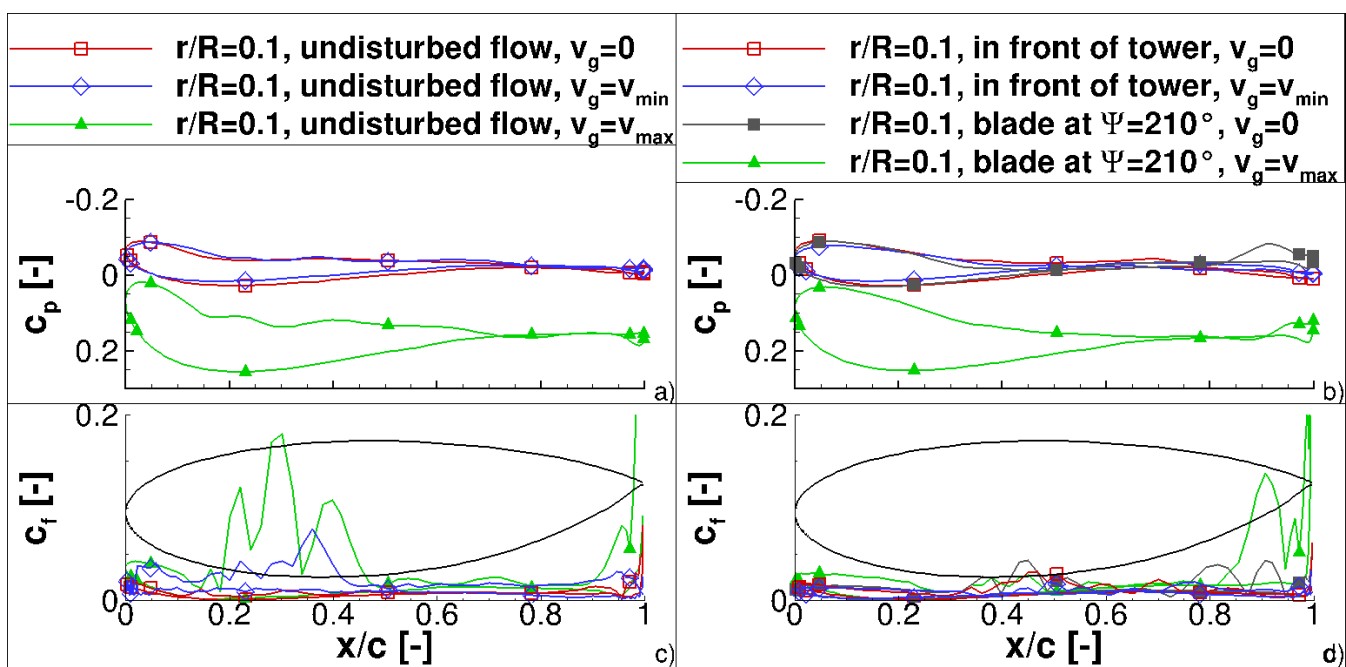

**Figure 8.** Pressure distribution of blade at inboard section; a,c) undisturbed flow; b,d) with tower blockage; a,b) pressure distribution, c,d) friction force coefficient

figure 8, 9 and 10, respectively.

In all three figures the pressure is displayed in the upper half and is normalized with the vector sum of the tip speed and the constant inflow velocity. For a meaningful comparison to constant inflow conditions, it was ensured that the investigated sections result from blades at the same azimuth positions.

A noticeable difference in $c_p$ at minimum gust velocity is found in all sections (figures 8a), 9a), and 10a)) when compared to the constant inflow conditions. The decrease in the stagnation point of $c_p$ to lower values at $t_{min}$ is higher than that of the tower blockage effect while otherwise the pressure distributions keep the general shape. Conversely the difference between constant inflow conditions and the maximum gust is significantly higher. The maximum $c_p$ increased about $58\%$ at the blade tip. Moreover, the shape of the pressure distribution changes over the entire blade. This is visible especially in the mid- and

outboard blade sections (figures 9a) and 10a)). In both sections, the pressure increases rapidly in the rear half of the upper blade surface and even reaches positive values in the last $20\%$ of the profile. This behaviour is a first indication of a separation region and reversed flow around the trailing edge.

The friction coefficients on the blade sections in undisturbed flow are displayed in figures 8c), 9c), and 10c). In all sections, strong fluctuations are visible at the trailing edge which result from the truncated geometry at $x/c = 1$. The friction force co-

efficient $c_f$ in the inboard section (figure 8c)) shows large differences between all considered time instances. The oscillations around $x/c = 50\%$ at constant inflow conditions indicate a small separation region with otherwise attached flow. At minimum

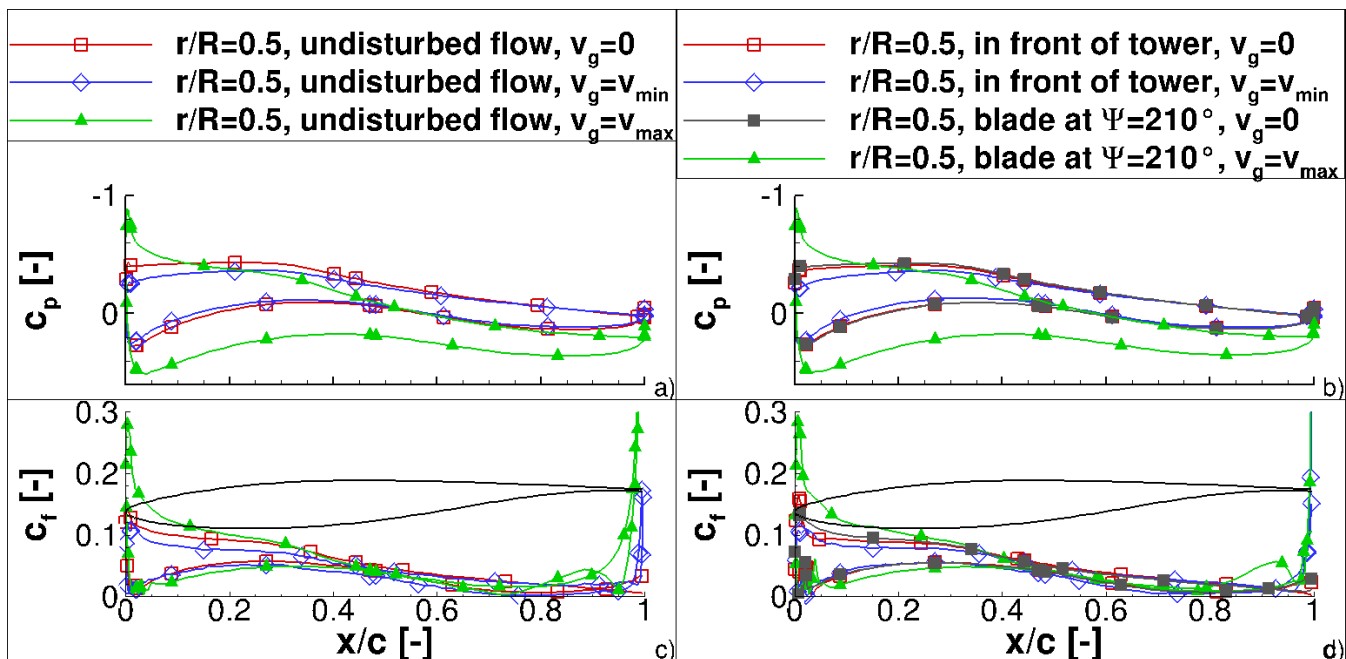

**Figure 9.** Pressure distribution of blade at mid section; a,c) undisturbed flow; b,d) with tower blockage; a,b) pressure distribution, c,d) friction force coefficient

gust velocity the overall friction force level is increased around the leading edge and the separation region has shifted upward and is between $x/c = 20\%$ and $x/c = 40\%$. At the maximum gust velocity, the friction force is increased and the oscillations between $x/c = 20\%$ and $x/c = 50\%$ indicate a larger separation region on the upper blade surface. The friction force coefficient indicates that separation in the blade inboard section is present during the entire rotor rotation. It is triggered through the

close cylindrical blade root and amplified with higher inflow velocities.

Conversely to the inboard section, changes in separation at the mid-section appear due to the gust only. In figure 9c) the friction force level is decreased at minimum gust velocity and the curve is very smooth. At maximum gust velocity, small oscillations appear around $x/c = 90\%$, indicating separation in that region. By increasing the rotor radius, the behaviour is enforced. At the outboard section (figure 10c)), the local maximum in $c_f$ in the last $20\%$ of the profile almost reaches the level of the leading

edge at $t_{max}$.

The same analysis is performed for the blade that is situated right in front of the tower or at $\Psi = 210°$. For the pressure distributions (figures 8b), 9b), 10b)) the same effects as described for the blades in undisturbed flow are found. The only difference is that due to the tower blockage the overall pressure level is decreased about $1\%$. This enforces the observation from the rotor thrust and rotor torque time histories in figure 5 and 7. Namely, the tower blockage effect is small compared to the

EOG operating loads. Conversely, the characteristics of friction forces (figures 8d), 9d), and 10d)) in the inboard and outboard sections differ from those of the blades in undisturbed flow. In the inboard section in figure 8d), the oscillations in $c_f$ reduce

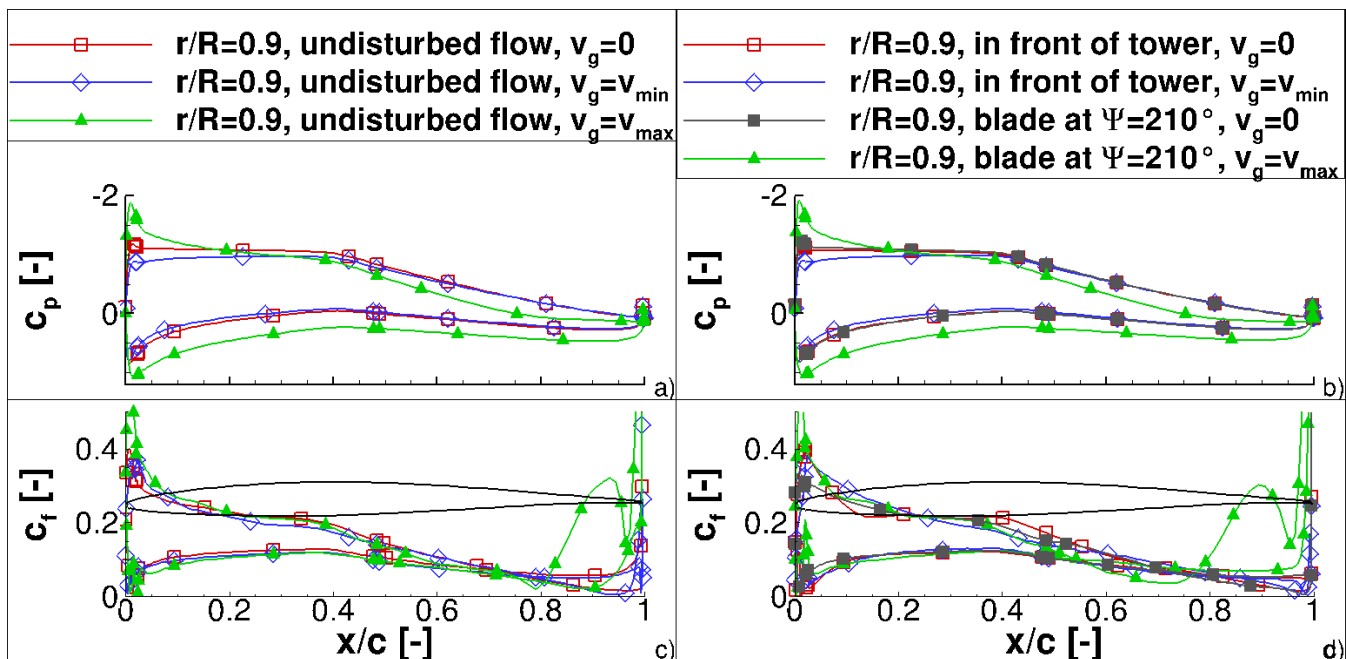

**Figure 10.** Pressure distribution of blade at outboard section; a,c) undisturbed flow; b,d) with tower blockage; a,b) pressure distribution, c,d) friction force coefficient

to a minimum while the overall friction force level remains constant. Only at maximum gust velocity, the oscillations around the trailing edge appear. Thus, the tower seems to suppress separation and it takes some time until the separation state is fully recovered. The friction force coefficient in the mid-section behaves similar to the inboard section with the difference that the entire separation region is larger (figure 9d)). In the outboard section in figure 10d), the friction force is increased significantly
due to the maximum gust velocity. By comparing the friction coefficients of the blades in undisturbed flow and in the tower blockage region, it is found that the separation on the suction side of the blade covers a larger area in the rear part of the blade. The separation induces a larger profile thickness which results to different induced angle of attack and thus reduced momentum.

Finally, the transport of the tip vortices is investigated. It has to be understood as indication whether the velocity transport in
the field works as expected but the tip-vortex transport has only small meaning on the transient rotor loading during the gust. Besides, the velocity in the field changes gradually because of the infinite speed of sound in the entire flow domain. Thus, the vortices that are shed from the blade at a given wind speed are not transported with their specific gust transport velocity. Contrariwise, all existing vortices experience identical changes in the gust transport velocity. Thus, the geometrical distance between existing vortices remains constant.

In figure 11 three instances of the flow field are compared. In all three instances, the rotor is at $\Psi = 0°$ positions. The vortices are made visible with the $\lambda_2$ criterion (Jeong and Hussain, 1995). The black lines are extracted during constant inflow, the red

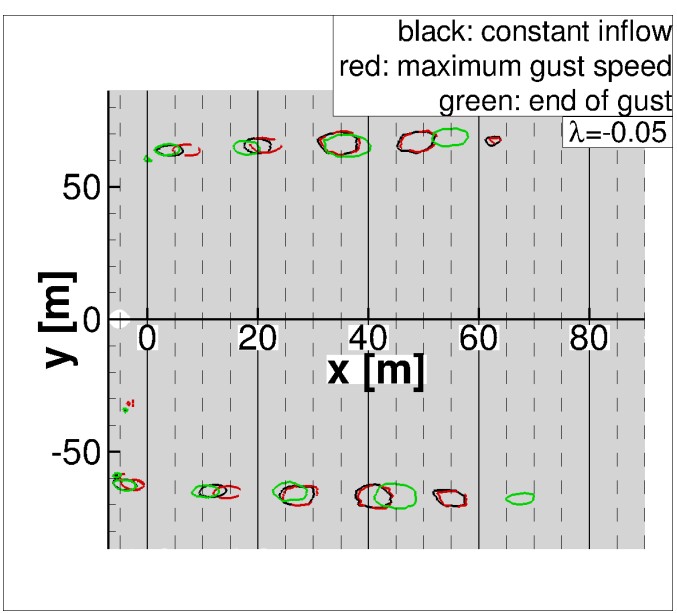

**Figure 11.** Tip vortex transportation in vertical plane through rotor centre

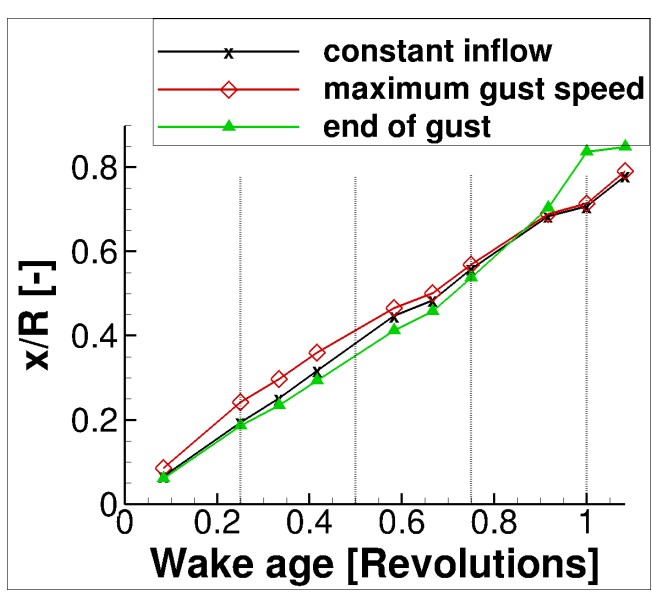

**Figure 12.** Tip vortex transportation in main flow direction in dependency of the wake age at three time instances during the gust

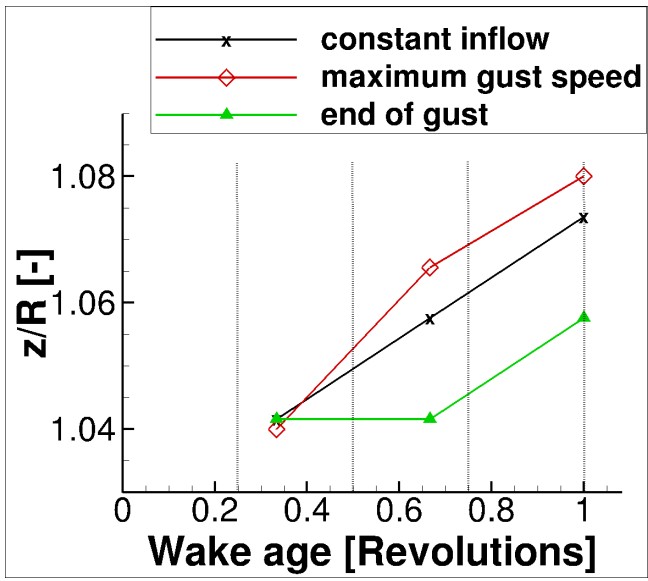

**Figure 13.** Tip vortex transportation vertical to main flow direction in dependency of the wake age at three time instances during the gust

lines are extracted shortly before $t_{max}$ and the green ones at the end of the gust. By comparing the vortex transport at the beginning of the gust (black curve) and at the end of the gust (green curve), a compression and stretching of the distance between the vortices is found. The vortex transport in dependency of the vortex age is further discussed through figures 12 and 13. They compare the transport of the vortex in dependency of the vortex age parallel and orthogonal to the flow direction, respectively.

5    As long as the inflow velocity is constant, the vortex transport is approximately $0.23 \cdot r/R$ parallel to the flow direction and $0.16 \cdot r/R$ orthogonal to the flow direction in a $1/3$-revolution interval. At maximum gust velocity, the constant distance between the vortices is lost. The vortices older than one revolution experienced constant inflow conditions. Thus the inter-vortex distance is constant. The vortices between $0.25$ and $1$ revolutions where shed during the reduced wind speed. Hence the distance parallel to the flow direction is reduced. As the downstream wake-transport decreases the orthogonal transport increases. 10    The vortices younger than $0.25$ revolutions experienced the high wind speed. Thus, the distance between the vortices parallel to the flow direction increases and the orthogonal transport decreases. At the end of the gust, reversed behaviour of the vortex transport is observed. The vortices between $0.75$ and $1$ revolution were generated while the wind speed slowed down. Thus, the distance between the vortices parallel to the wind direction is increased while the orthogonal transport is decreased. Vortices up to an age of $0.75$ revolutions have a decreased distance parallel to the wind direction but an increased distance orthogonal 15    to the wind direction.

The aerodynamic characteristics, rotor thrust and rotor torque of course depend on the assumption of stiff rotor blades and constant rotational speed. If the rotor had finite mass and inertia or a speed control algorithm had been applied, the rotor loading during the gust would have been reduced significantly. Moreover, the symmetry of the rotor loading decreases as soon as the structure dynamics are taken into account.

## 20   6   Conclusions

The study presented the validation of the resolved-gust approach that was implemented in the U-RANS-solver THETA. As test case, the generic $5\,\text{MW}$ wind turbine was computed, operating under a $1 - \cos()$-gust and an extreme operating gust as defined in the IEC 61400-1 (2005) standard. The gust has been introduced with the resolved-gust-approach (Kelleners and Heinrich, 2015; Reimer et al., 2015) by introducing the changing velocity at the inflow boundary conditions. The gust velocity 25    was then transported loss-free through the field with infinite speed of sound. The assumptions, which are made in the paper can be summarized as

     1. the wind speed is constant in height and time (except gust velocity)

     2. the gust velocity is constant in height

     3. the gust transport velocity is equal to speed of sound which is infinite

30      4. the boundary conditions of the flow domain are chosen to prevent the flow from escaping sideways.

The disadvantage of the resolved-gust approach clearly is the requirement of fine time steps and fine grids upstream of the geometry in question which increases the computational costs. The infinite speed of sound and its direct relation to the gust

transport velocity has to be regarded in its ambivalent effects. It leads to an inaccurate reproduction of the wind turbine wake transport on the one hand, while on the other hand, the response of the rotor loading to the gust velocity is immediately obtained. Clear advantages fo the resolved-gust approach are numerical stability and no artificial oscillations. Finally, the resolved-gust approach can be applied to any completed wind turbine U-RANS computation to obtain more insight of the wind turbine

characteristics.

The results represented the effects that are expected during the in-stationary inflow condition in combination with the given boundary conditions very well. Rotor thrust and rotor torque follow the gust shape very close. An analysis of the time history of rotor thrust and rotor torque during the gust show an increased rotor loading of about $100\,\%$ compared to constant inflow. Pressure distributions and friction force coefficients reveal that the flow on the rotor blades at maximum gust velocity is

separated and thus highly in-stationary. Moreover, the effect of accelerating wind speeds was found in the rotor wake as the distance between the vortices is stretched and compressed according to the changes of the wind speed.

The comparison of the results with the aero-elastic software FAST showed a very good agreement of rotor thrust and rotor torque during the EOG. Thus, it is a valid and accurate method to predict wind turbine loads during an EOG. Nevertheless, a complete validation is not possible at this state as a gust experiment for a wind turbine is not available. The first mandatory

step for further research on the gust simulation with U-RANS is to perform a grid-independence and time-step study with the resolved gust approach. Based on that results, a gust transport velocity with other than infinite speed of sound have to be achieved. This may either be realized by adjustments of the resolved-gust approach, by implementing the field approach of, for example, Parameswaran and Baeder (1997), or the velocity disturbance approach of Reimer et al. (2015). A third possibility would be to introduce the fluctuating gust velocities obtained from LES computations which themselves fulfil the continuity

conditions. Only then, the procedures of gust computation for wind turbines in THETA are prepared to be extended to respect atmospheric boundary layer flows or for aero-elastic analysis. The then ready-to-use method is supposed to supply a tool for gaining more detailed knowledge about the wind turbine behaviour during extreme gust events, in a first step. In a second step, this knowledge can be used to adjust engineering models which are used during the design process. It has to be clearly understood that the resolved-gust approach in a U-RANS computation can and will not replace engeeering models at this stage.

Consequently, the potential of weight reduction (and thus cost-redutction) and increased reliability in wind turbine designs are the (very-) long-term objectives.

*Acknowledgements.* The presented work was funded by the Federal Ministry of Economic Affairs and Energy of the Federal Republic of Germany under grant number 0325719.

*Data availability.* Availability of NREL $5\,\mathrm{MW}$ data from NREL reports; no other data available.

*Competing interests.* The author declares that she has no conflict of interest.

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
