# Peer review of "Simulation of transient gusts on the NREL5 MW wind turbine using the U-RANS-solver THETA"

_Wind Energy Science, 2017_

## Referee Comment (RC1) · Anonymous Referee #1 · 28 Nov 2017

Overall: The paper presents CFD(URANS) simulations of the NREL 5MW wind turbine. The aim is to examine transient gusts and the associated load implications. The underlying research question is interesting with some interesting conclusions, e.g. that the presence of the tower suppress separation. However, the article appears incomplete and somewhat unfocused, hence confusing at times. The written language is mediocre and could be improved for readability. Therefore, the recommendation is to reject the paper in the current state, but encourage to resubmit at a later stage. The author addresses a number of issues in the conclusion and these should be answered for resubmission. Hence, a more clear focus should be applied, e.g. how related is the tip vortex transportation actually to the transient loading during a gust? A number of suggestions will be given for improvements in the general comments, but detailed

comments are omitted.

General comments: 1. Why are the gusts propagated with the speed of sound? This appears an odd choice as a real gust would propagate by its own velocity. 2. Why is the floor modelled, but no ABL? If a no-slip condition is applied, why not have a shear? The dependance on height appears to disappear without explanation in the equtions. Otherwise, investigate a uniform inflow with the turbine "suspended" in space, which would give symmetry. 3. The turbine is stiff in the computations. Is this choice appropriate when examining extreme loads, particular for large wind turbines as the 5MW? This also affects the observed symmetry in Table 1 and whether this is to be expected. 4. Improved description of flow solver. What does dual cells, projection methods, prism layers, and C functions imply? 5. Why are there meshing issues involved for the nacelle, but not the hub nor tower? 6. How many cells in first grid? 7. The boundary condition definition is unclear, e.g. "the remaining farfield surface"? 8. How is such a small gust interesting? The cos gust basically results in a TI of 1.5%(=(sqrt(2)/2*0.25m/s)/11.4m/s, hardly a defining design case. 9. How are the characteristic times chosen? And why would they be sinusoidal? It might follow the standards, but does this correspond to measured gusts or gusts from LES? 10. Explain spikes in e.g. Figure 3 around t = 3230 sec. 11. Is the use of average loads correct? Most people use equivalent loads. 12. Details are difficult to see in Figures 7-12. 13. In terms of experiments, why not validate the setup against the MEXICO experiments or Krogstad as there is nothing "special" about the NREL 5MW.

Technical Corrections: 1. Why are all references written twice? Please correct. 2. Wording is often rather strange, e.g. use of "regarding", "promising"(page 3, line 27), "respecting", 3. Sentences are back-to-front, e.g. page 2, line 9-10. Please correct. 4. Consistency. On page 2, there are mentioned "CFD" several times, while aero-elastic tools are denominated. Please also specify which CFD tools were used as there are large differences between "CFD" tools. 5. Periodic and initial is not the same(page 7, line 15).

---

## Referee Comment (RC2) · Anonymous Referee #2 · 20 Dec 2017

Simulation of transient gusts on the NREL5 MW wind turbine using CFD

By Annika Länger-Möller

The article describes a CFD approach to gust modeling for wind turbines, presenting some interesting results.

As the CFD model is incompressible, any change in velocity in the domain is instantaneously influencing the full domain. As a consequence, the gust is not traveling through the domain, but the velocity is basically increasing everywhere in the domain instantaneously. The consequence of this with respect to the wake development should be discussed.

Additionally, the assumption of infinite mass and inertia along with a non-elastic model

might also have quite large effects, and should be discussed.

Finally, the ABL is neglected along with any turbulence, which is also indicated by other reviewers could be much larger than the cosines gust.

Figures:

Generally the figures are very small, and Figure 1 about the grid set-up is basically of no use and should be replaced. The Cp and Cf plots, Fig. 7,8,9,10, 11 and 12 are very small and not providing that much information. They could be exchanged with plots of radial force distributions at [0,90,180,270] degrees azimuth. Eventually, a blow-up of the Cf distribution could be included to assist the discussion about separation versus no separation.

In Figure 13, the choice of blue and black color is not optimal, red would be easier to distinguish from the black.

Flow Solver:

The description of the flow solver setup is very sparse and should be extended.

What time integration is used for the present work, and what size of time step is used.

It is stated that a second order central scheme is used, but it seems highly unrealistic that this can be done without generating wiggles for this high a Reynolds number without some artificial damping. Please explain.

The present reviewer is well aware of the no-slip wall conditions used, but I do not understand how it differs from the viscous wall condition prescribed at the earth surface. Should it be an inviscid wall or slip conditions at the earth surface?

Grid Characteristics

The description is not accurate and do not even include any description of the gridding of the tower component.

Additionally, some more details about the issues related to including the nacelle in the grid should be given.

The chord-wise resolution is on the coarse side; normally more than 250 cells are needed for an accurate resolution of the flow development. Are the results at all close to grid independent?

More illustrative figures showing the chord-wise and span-wise resolution should be included, along with a cut through the full grid topology.

Results:

Generally, the pressure distributions add very little information to the discussion, and should be replaced by span-wise force distributions.

Figure 13 on the tip vortex movement is very hard to interpret, as we do not know if the rotor is at identical azimuth positions for the different snap-shots.

I would maybe be more interesting to show the axial and radial location of the tip-vortex as function of vortex age at the three instances in time.

References:

The major references to CFD for wind turbines are very recent; CFD for wind turbines track back to the late nineties which I believe should be reflected.

Additionally, the author chose to refer to secondary references eg. Kessler and Lôwe 2012, where the reference to Zhang et al from my point of view should be preferred.

P3, L15 moving grid blocks (Zhang et al. 2007) as implemented by (Kessler and Lôwe 2012).

Another example is the reference for the usage of k-omega model for wind turbines which has been pioneered by others in the late nineties, e.g.

Rotor performance prediction using a Navier-Stokes Method, Sørensen and Hansen,

AIAA-98-0025.

---

## Author Comment (AC2) · 12 Jan 2018

Dear Dear Anonymous Referee #2,
thank you very much for your hints to improve the analysis of my results. I especially appreciated the recommendation of references. Furthermore I will answer your questions.
As I introduced a validation of the computation as response to Referee #1, section 2 now contains two subsections. The first one for THETA, the second to describe the simulation setup in FAST.

[Figure]

**Overall**

- RC: *The article describes a CFD approach to gust modelling for wind turbines, presenting some interesting results. As the CFD model is incompressible, any change in velocity in the domain is instantaneously influencing the full domain. As a consequence, the gust is not travelling through the domain, but the velocity is basically increasing everywhere in the domain instantaneously. The consequence of this with respect to the wake development should be discussed.*
  AC: Yes. Indeed, the entire vortex-transport discussion is turning around this point. To clarify this I added the paragraph "In consequence to the infinite speed of sound in the entire flow domain, the velocity in the field changes gradually. Thus, the vortices that are shed from the blade at a given wind speed are not transported with their specific transport velocity. Contrariwise, all existing vortices experience identical changes in the transport velocity. In consequence, the geometrical distance between existing vortices remains constant." to section 5.3.

- RC: *Additionally, the assumption of infinite mass and inertia along with a non-elastic model might also have quite large effects, and should be discussed.*
  AC: Yes you are right. A finite mass and inertia would reduce the rotor loading during the gust. Moreover, elastic deformation during the gust becomes important in the given test case and would destroy the symmetry in the rotor loading. These trends can be seen by switching on the according models in FAST. Nevertheless, the motivation of the paper has been to validate the resolved gust approach in THETA. In this context it is best to diminish the uncertainties by reducing the model complexity. This is done by isolating the aerodynamic effects. The discussion of results is enhanced by the information to trends for finite mass and inertia (section 5.2, 5.3). Section 1 now contains a more precise definition of the scope of the paper.

- RC: *Finally, the ABL is neglected along with any turbulence, which is also indi-*

*cated by other reviewers could be much larger than the cosines gust.*

AC: The cosines-gust represents the turbulence level of the atmospheric flow around wind turbines as measured by Schaffarczyk et al. (2017). A remark is added at the end of section 4.2. The ABL inflow profile can be added after the resolved-gust approach has been validated successfully. This point is clarified in the paper in section 3.1, 4.1, 4.3.

**Figures**

- RC: *Generally the figures are very small, and Figure 1 about the grid set-up is basically of no use and should be replaced.*

  AC: I enlarged the figure. Moreover I changed the content to represent the chord-wise distribution, span-wise distribution in the blade tip region and the meshing topology in the farfield. Please see figure 1 of my response.

- RC: *The Cp and Cf plots, Fig. 7,8,9,10, 11 and 12 are very small and not providing that much information.*

  AC: I enlarged figure 7 to 12 and summarized them into 3 figures to enable a better comparison between no tower/tower blockage. I added one exemplary figure to my response. The descriptive text in the paper changed accordingly. Please see figure 2 of my response.

- RC: *They could be exchanged with plots of radial force distributions at [0,90,180,270] degrees azimuth. Eventually, a blow-up of the Cf distribution could be included to assist the discussion about separation versus no separation.*

  AC: I added a plot of the radial distribution of rotor thrust and torque to the paper. Please refer to section Results, bullet-point 1 for further information.
- RC: *In Figure 13, the choice of blue and black color is not optimal, red would be easier to distinguish from the black.*
  AC: I changed the colouring from blue to red.

**Flow solver**

- RC: *The description of the flow solver setup is very sparse and should be extended. What time integration is used for the present work, and what size of time step is used.*
  AC: Eulerian Implicit time stepping scheme, global time stepping. The information is added. A time step of $0.006887052s$ is used which is equivalent to a rotor advance of $\delta\Psi = 0.5°$. The information has partly been at the end of section 2 but is expanded.

- RC: *It is stated that a second order central scheme is used, but it seems highly unrealistic that this can be done without generating wiggles for this high a Reynolds number without some artificial damping. Please explain.*
  AC: Please refer to page 3, line 13: "Pressure stabilization is used to avoid spurious oscillations caused by the collocated variable arrangement." This pressure stabilization prevents the central scheme from oscillating. No further comment is made in the paper.

- RC: *The present reviewer is well aware of the no-slip wall conditions used, but I do not understand how it differs from the viscous wall condition prescribed at the earth surface. Should it be an inviscid wall or slip conditions at the earth surface?*
  AC: It has been a typing error in page 4, line 22. The floor is a viscous wall but the top, left, and right surface of the flow domain are slip walls. I changed the word no-slip to slip in the given line.

**Grid characteristics**

- RC: *The description is not accurate and do not even include any description of the gridding of the tower component.*
  AC: The tower surface grid is meshed structured in a height below $5\,\mathrm{m}$ with 54 point in height and 180 points in radial direction. Above this grid, a triangulated unstructured grid is generated. The maximum edge length is $0.55\,\mathrm{m}$. As the tower surface is modelled as slip-wall, tetrahedrons are built directly on the tower surface. This information is added to section 3.2.

- RC: *Additionally, some more details about the issues related to including the nacelle in the grid should be given.*
  AC: Due to the narrow gap between rotor and nacelle a valid chimera overlap-region could not be achieved. Thus the nacelle of the NREL 5MW turbine is neglected while the tower is respected. This sentence is added to section 3.2

- RC: *The chord-wise resolution is on the coarse side; normally more than 250 cells are needed for an accurate resolution of the flow development. Are the results at all close to grid independent?*
  AC: In Länger-Möller (2017) it has been shown that with even coarser distribution in chord-wise direction, the results of THETA matched the NREL UAE phase VI perfectly. Based on this findings the mesh of the NREL 5MW turbine was generated. Moreover, the very good agreement to the NREL 5MW documentation of Jonkman et al. (2009) (section 5.1) indicate that the grid is fine enough.

- RC: *More illustrative figures showing the chord-wise and span-wise resolution should be included, along with a cut through the full grid topology.*
  AC: See section Figures, first bullet point.
**Results**

- RC: *Generally, the pressure distributions add very little information to the discussion, and should be replaced by span-wise force distributions.*
  AC: Thank you for the suggestion. I prepared figures of the span-wise force distribution and added them to the paper. It is also attached to my reply. In the paper, a new paragraph is added to section 5.3. Please see figure 1 of my response.

- RC: *Figure 13 on the tip vortex movement is very hard to interpret, as we do not know if the rotor is at identical azimuth positions for the different snap-shots.*
  AC: Yes, the instances were taken at identical azimuth positions. This information is added to section 5.3.

- RC: *It would maybe be more interesting to show the axial and radial location of the tip-vortex as function of vortex age at the three instances in time.*
  AC: I changed figure 14 accordingly to your suggestions and added a plot of the axial and radial location of the tip vortex as function of vortex age. Conversely, I only changed the colouring of figure 13 from blue to red. I attached the figures to give you an impression. The description of the figures changed accordingly in section 5.3. Please see figure 4 of my response.

**References**

- RC: *The major references to CFD for wind turbines are very recent; CFD for wind turbines track back to the late nineties which I believe should be reflected.*
  AC: I summarized the milestones in CFD for wind turbines in some additional sentences right at the beginning of section 1. I namely refer to Soerensen and Hansen (1998), Soerensen et al. (2002), Johansen et al. (2002), Bazilevs et al. (2011), Hsu et al. (2012), and Chow and van Dam (2012).

- RC: *Additionally, the author chose to refer to secondary references eg. Kessler and Löwe 2012, where the reference to Zhang et al from my point of view should be preferred. P3, L15 moving grid blocks (Zhang et al. 2007) as implemented by (Kessler and Löwe 2012).*
  AC: I followed your suggestion and additionally added a reference to Pan et al. (2002). The information is added to section 2.1.

- RC: *Another example is the reference for the usage of k-omega model for wind turbines which has been pioneered by others in the late nineties, e.g. Rotor performance prediction using a Navier-Stokes Method, Sørensen and Hansen, AIAA-98-0025.*
  AC: I never intended to state that Länger-Möller pioneered the Menter SST model for wind turbine applications. Anyway, I clarified that during the validation of THETA by Länger-Möller et al. (2017) the Menter-SST model has been the most accurate turbulence model. Thus it is used in the present study. The information will be given in section 2.1.
* * *
[Figure]

**Fig. 1.** Grid resolution

[Figure]

**Fig. 2.** pressure coefficient and friction force coefficient

**Fig. 3.** radial distribution of rotor thrust and torque

**Fig. 4.** vortex transport parallel to flow direction in dependency of vortex age

[Figure]

---

## Author Response (AR1)

**Dear Anonymous Referee #1,**

Thank you very much for your comments that certainly will improve the paper. In the following I will answer your questions and indicate which information is added to the original paper.

**Overall**

5   RC:*The paper presents CFD(URANS) simulations of the NREL 5MW wind turbine. The aim is to examine transient gusts and the associated load implications. The underlying research question is interesting with some interesting conclusions, e.g. that the presence of the tower suppress separation. However, the article appears incomplete and somewhat unfocused, hence confusing at times. The written language is mediocre and could be improved for readability. Therefore, the recommendation is to reject the paper in the current state, but encourage to resubmit at a later stage. The author addresses a number of issues in the*

10   *conclusion and these should be answered for resubmission. Hence, a more clear focus should be applied, e.g. how related is the tip vortex transportation actually to the transient loading during a gust?*

AC: To improve the language, I shortened the sentences and reordered them to strictly match the subject-verb-object order. Moreover, I changed some wordings to improve the readability of the paper. This hopefully will clarify the focus on the vali-

15   dation of the method, which is the main purpose of the paper.
I also fixed the main issue in my conclusion, namely the validation of the resolved-gust approach. The CFD(U-RANS)-computation is now compared to FAST and shows very good agreement in flow regions with attached flow. I added the according figure to my response. Besides, everything else would mean to try to run before I can walk.

[Figure]

20   The tip vortex transport is an indication whether the method works correctly. It has little effect on the rotor loading. This information is added to section 5.3.

**General Comments**

1. RC: *Why are the gusts propagated with the speed of sound? This appears an odd choice as a real gust would propagate by its own velocity.*

25   RC: Yes you are right. It is the weak point of the chosen modelling approach. In combination with an incompressible solver it becomes severe as the speed of sound is infinite. This point has already mentioned in section 1, page 2, line 27. The use of the field approach by Parameswaran et al. or the velocity disturbance approach would overcome this issue but introduce other problematic questions. I will expand the explanation in section 1.

2. RC:*Why is the floor modelled, but no ABL? If a no-slip condition is applied, why not have a shear? The dependence on height appears to disappear without explanation in the equations. Otherwise, investigate a uniform inflow with the turbine "suspended" in space, which would give symmetry.*

   AC: To be able to introduce the ABL in future computations and to perform a comparison of both computations on the same grid, the grid has already been prepared accordingly. I added this information to section 3.2. Nevertheless, the first step in introducing a new method is to validate the simulation method as isolated as possible. Thus, the height dependency of the velocity is neglected to be able to evaluate the gust impact on the rotor aerodynamics. I added the information more clearly in the paper in section 3.1 4.1, 4.3.

3. RC: *The turbine is stiff in the computations. Is this choice appropriate when examining extreme loads, particular for large wind turbines as the 5MW? This also affects the observed symmetry in Table 1 and whether this is to be expected.*

   AC: The first step in introducing a new method with interdisciplinary character is to validate each part of the method on its own. The combination with structure dynamics and/or speed controllers can only follow after the issues in remark [1] and [13] are clarified. The motivation is clarified in section 1, page 2. Moreover, a remark about the plausibility of symmetric results is added in section 5.2, 5.3.

4. RC:*Improved description of flow solver. What does dual cells, projection methods, prism layers, and C functions imply?*

   AC: Dual cells imply a cell-centred scheme of the solver. With the projection method the momentum equations are first solved with an approximated pressure field and do not fulfil continuity. The pressure field is then corrected by solving a Poisson equation to fulfil the continuity conditions. The c-functions enable a high flexibility on the definition of inflow conditions on the boundaries of the flow domain. The description of THETA in section 2 is extended to answer your questions. Prism layers are part of the meshing topology.

5. RC: *Why are there meshing issues involved for the nacelle, but not the hub nor tower?*

   AC: Due to the narrow gap between rotor and nacelle a valid chimera overlap-region could not be achieved. Thus the nacelle of the NREL 5MW turbine is neglected while the tower is respected. This sentence is added to section 3.2

6. RC:*How many cells in first grid?*

   AC: 11.6 Mio in the chimera child grid and 13.3 Mio points in the parent grid. The information has been already on page 4, lines 20 and 26 in the first version.

7. RC:*The boundary condition definition is unclear, e.g. "the remaining farfield surface"?*

   AC: It is the surfaces on top, left and right of the flow domain. I clarified this in the paper, section 3.2.

8. RC: *How is such a small gust interesting? The cos gust basically results in a TI of $1.5\%$ (=(sqrt(2)/2*0.25m/s)/11.4m/s, hardly a defining design case.*

   AC: It represents the turbulence level of the atmospheric flow around wind turbines as measured by Schaffarczyk et al. (2017). A remark is added at the end of section 4.2.

9. RC: *How are the characteristic times chosen? And why would they be sinusoidal?*

   AC: The characteristic time for the 1-cos gust was chosen to generate an un-compressed cosinus larger than one rotor revolution and smaller than a 10s interval. The characteristic time for the EOG is taken from the IEC standard. A comment is added in the sections 4.2 and 4.3.

9. RC: *And why would they be sinusoidal? It might follow the standards, but does this correspond to measured gusts or gusts from LES?*

   AC: Well, this opens up a wide research field and several attempts are made to find profound answers to this question. One example is Bierbooms et al. (1999) who investigated the gust shape from wind measurements and simulation. By the use of statistical methods they found that a sinus fairly represents gust. Alternatively, Zbrozek (1953) gives an overview on different possible gust shapes. Concerning the agreement of the standards and results from LES or field measurements, Mücke et al. (2010), for example, showed that in field measurements velocity-changes with high frequency occur. These are not represented in the IEC standard. Apart from that, the question to be answered by my

paper neither is whether or not the implemented gust corresponds to any field measurement or LES computation with the specific related uncertainties. Nor it is the purpose to answer the question how the IEC standard may be improved. The aim of the paper is to validate the resolved gust approach, which requires

- low uncertainties in the inflow conditions which is achieved by perfect control of the inflow conditions
- low uncertainties in the inflow conditions by applying a gust that is independent of the horizontal and vertical position
- comparability to results of other methods such as FAST by using well-defined inflow conditions

All of the above requirements are best matched by the IEC standard gust. The introduction of realistic fluctuations which result from LES or measurement can be applied as soon as the validation of the aerodynamics is finished, aero-elasticity has been included and a speed controller is available.

10. RC: *Explain spikes in e.g. Figure 3 around t = 3230 sec.*
    AC: The high-frequency oscillations occur in a time step wherein the elliptic pressure equation has not reached the required residual of $10^{-5}$ in the maximum allowed number of iterations. The number of iterations were chosen to guarantee an efficient computation of the test case. The information about the criteria is added to section 2. Moreover, a remark on the high-frequency oscillations is added to section 5.1.

11. RC: *Is the use of average loads correct? Most people use equivalent loads.*
    AC: To be able to compute and discuss equivalent loads assumptions on the wind field, wind velocity bins and statistics have to be included to the discussion in the paper. This would create new questions and unnecessary uncertainties to the results. Therefore I included a more profound comparison of averaged and peak loads to section 5.2 that clarifies the introduction of the averaged gust loads.

12. RC: *Details are difficult to see in Figures 7-12.*
    AC: I modified the figures to better illustrate the content. Moreover, I enlarged the figures.

13. RC: *In terms of experiments, why not validate the setup against the MEXICO experiments or Krogstad as there is nothing "special" about the NREL 5MW.*
    AC: The validation of the setup against any experiment is not a matter of question because a validation against the NREL 5MW is performed in section 5.1 and a successful validation of THETA against the NREL phase VI UAE has been presented by Länger-Möller in 2017. It is the gust simulation that needs validation. Meanwhile I found a possibility to validate the gust simulation against FAST results. I included the results to figure 5 and according comments to section 5.3. The Conclusion in section 6 has been changed accordingly.

14. I added a short description of FAST and the modelling approaches in section 2. That includes a modification of the section title to "Numerical methods" and a shift of the former section 2 to 2.1

**Technical corrections:**

1. RC: *Why are all references written twice? Please correct.*
   AC: References are now written once.

2. RC: *Wording is often rather strange, e.g. use of "regarding", "promising"(page 3, line 27), "respecting",*
   AC: I changed the wording you indicated.

3. RC: *Sentences are back-to-front, e.g. page 2, line 9-10. Please correct*
   AC: I changed the wording in sentences on page 2, line 9-10 and some other back-to-front sentences.

4. RC: . *Consistency. On page 2, there are mentioned "CFD" several times, while aero-elastic tools are denominated. Please also specify which CFD tools were used as there are large differences between "CFD" tools.*

   AC: I specified the used CFD tools more clearly. This implies the title. Moreover I added a short description of the most popular aero-elastic wind turbine tools.

5. RC: *Periodic and initial is not the same(page 7, line 15).*

   AC: Of course, periodic and initial is not the same. But the periodic state of the flow field is mandatory as starting (initial) conditions to start a gust-computation. I clarified this in the mentioned paragraph.

Moreover, I performed the following changes:

– I changed the word "hexahedra" to "hexagon"

– I adapted wind velocity to wind speed

– I adapted gust speed to gust velocity

throughout the entire paper.

**Dear Dear Anonymous Referee #2,**

thank you very much for your hints to improve the analysis of my results. I especially appreciated the recommendation of references. Furthermore I will answer your questions.

As I introduced a validation of the computation as response to Referee #1, section 2 now contains two subsections. The first
5  one for THETA, the second to describe the simulation setup in FAST.

**Overall**

– RC: *The article describes a CFD approach to gust modelling for wind turbines, presenting some interesting results. As the CFD model is incompressible, any change in velocity in the domain is instantaneously influencing the full domain. As a consequence, the gust is not travelling through the domain, but the velocity is basically increasing everywhere in*
10      *the domain instantaneously. The consequence of this with respect to the wake development should be discussed.*
      AC: Yes. Indeed, the entire vortex-transport discussion is turning around this point. To clarify this I added the paragraph "In consequence to the infinite speed of sound in the entire flow domain, the velocity in the field changes gradually. Thus, the vortices that are shed from the blade at a given wind speed are not transported with their specific transport velocity. Contrariwise, all existing vortices experience identical changes in the transport velocity. In consequence, the geometrical
15      distance between existing vortices remains constant." to section 5.3.

– RC: *Additionally, the assumption of infinite mass and inertia along with a non-elastic model might also have quite large effects, and should be discussed.*
      AC: Yes you are right. A finite mass and inertia would reduce the rotor loading during the gust. Moreover, elastic deformation during the gust becomes important in the given test case and would destroy the symmetry in the rotor
20      loading. These trends can be seen by switching on the according models in FAST. Nevertheless, the motivation of the paper has been to validate the resolved gust approach in THETA. In this context it is best to diminish the uncertainties by reducing the model complexity. This is done by isolating the aerodynamic effects. The discussion of results is enhanced by the information to trends for finite mass and inertia (section 5.2, 5.3). Section 1 now contains a more precise definition of the scope of the paper.

25  – RC: *Finally, the ABL is neglected along with any turbulence, which is also indicated by other reviewers could be much larger than the cosines gust.*
      AC: The cosines-gust represents the turbulence level of the atmospheric flow around wind turbines as measured by Schaffarczyk et al. (2017). A remark is added at the end of section 4.2. The ABL inflow profile can be added after the resolved-gust approach has been validated successfully. This point is clarified in the paper in section 3.1, 4.1, 4.3.

30  **Figures**

– RC: *Generally the figures are very small, and Figure 1 about the grid set-up is basically of no use and should be replaced.*
      AC: I enlarged the figure. Moreover I changed the content to represent the chord-wise distribution, span-wise distribution

in the blade tip region and the meshing topology in the farfield. I attached the figure to my response.

[Figure]

– RC: *The Cp and Cf plots, Fig. 7,8,9,10, 11 and 12 are very small and not providing that much information.*
AC: I enlarged figure 7 to 12 and summarized them into 3 figures to enable a better comparison between no tower/tower blockage. I added one exemplary figure to my response. The descriptive text in the paper changed accordingly.

[Figure]

– RC: *They could be exchanged with plots of radial force distributions at [0,90,180,270] degrees azimuth. Eventually, a blow-up of the Cf distribution could be included to assist the discussion about separation versus no separation.*
AC: I added a plot of the radial distribution of rotor thrust and torque to the paper. Please refer to section Results, bullet-point 1 for further information.

– RC: *In Figure 13, the choice of blue and black color is not optimal, red would be easier to distinguish from the black.*
AC: I changed the colouring from blue to red.

**Flow solver**

– RC: *The description of the flow solver setup is very sparse and should be extended. What time integration is used for the present work, and what size of time step is used.*
AC: Eulerian Implicit time stepping scheme, global time stepping. The information is added. A time step of $0.006887052s$ is used which is equivalent to a rotor advance of $\delta\Psi = 0.5°$. The information has partly been at the end of section 2 but is expanded.

– RC: *It is stated that a second order central scheme is used, but it seems highly unrealistic that this can be done without generating wiggles for this high a Reynolds number without some artificial damping. Please explain.*
AC: Please refer to page 3, line 13: "Pressure stabilization is used to avoid spurious oscillations caused by the collocated variable arrangement." This pressure stabilization prevents the central scheme from oscillating. No further comment is made in the paper.

– RC: *The present reviewer is well aware of the no-slip wall conditions used, but I do not understand how it differs from the viscous wall condition prescribed at the earth surface. Should it be an inviscid wall or slip conditions at the earth surface?*
AC: It has been a typing error in page 4, line 22. The floor is a viscous wall but the top, left, and right surface of the flow domain are slip walls. I changed the word no-slip to slip in the given line.

**Grid characteristics**

– RC: *The description is not accurate and do not even include any description of the gridding of the tower component.*
AC: The tower surface grid is meshed structured in a height below $5\,\mathrm{m}$ with 54 point in height and 180 points in radial direction. Above this grid, a triangulated unstructured grid is generated. The maximum edge length is $0.55\,\mathrm{m}$. As the tower surface is modelled as slip-wall, tetrahedrons are built directly on the tower surface. This information is added to section 3.2.

– RC: *Additionally, some more details about the issues related to including the nacelle in the grid should be given.*
AC: Due to the narrow gap between rotor and nacelle a valid chimera overlap-region could not be achieved. Thus the nacelle of the NREL 5MW turbine is neglected while the tower is respected. This sentence is added to section 3.2

– RC: *The chord-wise resolution is on the coarse side; normally more than 250 cells are needed for an accurate resolution of the flow development. Are the results at all close to grid independent?*
AC: In Länger-Möller (2017) it has been shown that with even coarser distribution in chord-wise direction, the results of THETA matched the NREL UAE phase VI perfectly. Based on this findings the mesh of the NREL 5MW turbine was generated. Moreover, the very good agreement to the NREL 5MW documentation of Jonkman et al. (2009) (section 5.1) indicate that the grid is fine enough.

– RC: *More illustrative figures showing the chord-wise and span-wise resolution should be included, along with a cut through the full grid topology.*
AC: See section Figures, first bullet point.

**Results**

– RC: *Generally, the pressure distributions add very little information to the discussion, and should be replaced by span-wise force distributions.*
AC: Thank you for the suggestion. I prepared figures of the span-wise force distribution and added them to the paper. It

is also attached to my reply. In the paper, a new paragraph is added to section 5.3.

[Figure]

- RC: *Figure 13 on the tip vortex movement is very hard to interpret, as we do not know if the rotor is at identical azimuth positions for the different snap-shots.*
  AC: Yes, the instances were taken at identical azimuth positions. This information is added to section 5.3.

- RC: *It would maybe be more interesting to show the axial and radial location of the tip-vortex as function of vortex age at the three instances in time.*
  AC: I changed figure 14 accordingly to your suggestions and added a plot of the axial and radial location of the tip vortex as function of vortex age. Conversely, I only changed the colouring of figure 13 from blue to red. I attached the figures to give you an impression. The description of the figures changed accordingly in section 5.3.

[Figure]

**List of changes to the original paper**

To improve the original paper, the following changes have been made:

**Modification of Text**

- I corrected the language wherever it was necessary (recommendation by Anonymous Reviewer #1)

- I changed the word "hexahedra" to "hexagon"

- I adapted wind velocity to wind speed

- I adapted gust speed to gust velocity

- I corrected the display of citations (requested by Anonymous Reviewer #1)

- I added a description of older CFD literature in the preamble (suggestion by Anonymous Reviewer #2)

- I added a description of the aero-elastic tools for the computation of wind turbines (suggestion by Anonymous Reviewer #1)

- I formulated the purpose of the paper more precisely (requested by Anonymous Reviewer #1)

- I added a description of different gust modelling approaches in U-RANS solvers (suggestion by Anonymous Reviewer #1)

- I now distinguish between U-RANS and LES (suggestion by Anonymous Reviewer #1)

- I extended the description of the flow solver and the solver settings during the computation (suggestion by Anonymous Reviewer #1 and #2)

- I added a description of FAST which is needed for the validation (requested by Anonymous Reviewer #1)

- I added a description of the tower grid (requested by Anonymous Reviewer #2)

- I changed the text in section 3.2 to match the changes in figure 1

- I added a computation of the turbulence level in the $1 - \cos()$-gust case and gave a reason for the choice (suggestion by Anonymous Reviewer #1 and #2)

- I explained the oscillations in rotor thrust and rotor torque in figure 3,4 (suggestion by Anonymous Reviewer #1)

- I commented on the symmetric response to the gust in connection with the modelling assumption (suggestion by Anonymous Reviewer #1)

- I added a comparison between the THETA computation and FAST for validation (suggestion by Anonymous Reviewer #1)

- I added a description of the newly introduced figure 7 and clarified the information about the blade position of the investigated time instances (requested by Anonymous Reviewer #2)

- I added a comment about the link between rotor loading and vortex transport (suggestion by Anonymous Reviewer #1)

- I reflected on the dependence of infinite speed of sound and vortex transport in the case of changing wind speed (suggestion by Anonymous Reviewer #2)

- I deleted the description of figure 14 (suggestion by Anonymous Reviewer #2)

- I added a discussion for the newly introduced figures 12, 13. This implies a reduction of the discussion of the former figure 13, now figure 11 (suggested by Anonymous Reviewer #2)

- I changed the conclusion to match the changes in the paper.

**Modifications of Figures**

- I enlarged the figures

- I modified figure 1, to better represent the grid resolution

- I modified figure 5 to additionally include the validation with FAST

- I introduced figure 7 to display the span-wise distribution of rotor thrust and rotor torque (suggestion by Anonymous Reviewer #2)

- I summarized figures 7 to 11 to figure 8-10 to clarify the results (requested by Anonymous Reviewer #1 and #2)

- I changed the colour maps of figure 13 (now 11) (suggestion by Anonymous Reviewer #2)

- I deleted figure 14. Instead I entered 2 figures which are better suited to forward the discussion of vortex transport. (suggestion by Anonymous Reviewer #2)

**Simulation of transient gusts on the NREL5 MW wind turbine using the U-RANS-solver THETA**

Annika Länger-Möller[1]

[1]DLR Braunschweig; Lilienthalplatz 7; 38108 Braunschweig

*Correspondence to:* Annika Länger (annika.laenger@dlr.de)

**Abstract.**  A procedure to propagate longitudinal transient gusts through a flow field by using  the resolved-gust approach is implemented in the U-RANS solver THETA. Both, the gust strike of a $1 - \cos()$-gust and an extreme operating gust following the IEC 61400-1 standard are investigated, on the generic NREL 5MW wind turbine at rated operating conditions
. The impact of both gusts on pressure distributions, rotor thrust, rotor torque, and flow states on the blade are examined and quantified. The flow states on the rotor blade before the gust strike at maximum and minimum gust velocity are compared. An increased blade loading is detectable in the pressure coefficients and integrated blade loads. The friction force coefficients indicate the dynamic separation and re-attachment of the flow during the gust. Moreover, a validation of the method is performed by comparing the rotor torque during the extreme operating gust to results of FAST rotor code.

*Copyright statement.* The works published in this journal are distributed under the Creative Commons Attribution 3.0 License. This licence does not affect the Crown copyright work, which is re-usable under the Open Government Licence (OGL). The Creative Commons Attributions 3.0 License and the OGL are interoperable and do not conflict with, reduce or limit each other.

[revised manuscript text omitted]

---

## Referee Report (RR1)

**Review of 'Simulation of transient gusts on the NREL 5 MW wind turbine using the U-RANS-solver THETA'**
**by Annika Länger-Möller**

The article presents the results of CFD simulations for a wind turbine model under transient gust conditions. The NREL 5 MW wind turbine with rotor and tower only is modelled. The CFD solver used is called U-RANS solver THETA and it is an in-house incompressible NS code developed in DLR. A resolved-gust approach, which is introducing the changing velocity at the inflow boundary conditions ensuring the loss-free transport of gust velocity in the flow field, is implemented in the CFD solver to simulate transient gust conditions based on the IEC 61400-1 standard. Unstructured overset grids are used in the simulations. Comparisons of the CFD results are made with the results of FAST analysis for validation of the new approach.

Comments:

1. Please comment and discuss on followings to clarify the implementation of the resolved gust approach in an incompressible solver:

a. Loss-free transport of gust velocity through the flow field may require the free-stream velocity update not only on inflow boundary but on other far boundaries (top and sides) as well, or actually whole domain if necessary. How are the slip-condition velocities (or frestream conditions changing on these far boundaries? Gust model is changing inflow velocity i.e., free-stream, and inflow velocity will effect all boundaries. Please clarify.

b. And no-slip condition on the bottom boundary will result in somehow development of a Boundary Layer (BL) in the unsteady solution. Is there any development and effect of this BL on the tower and rotor, or flow field? Please clarify.

2. Discussing clearly all the assumptions and difficulties encountered in such expensive unsteady computations will be helpful for future studies.

---

## Author Response (AR2)

**Dear Anonymous Referee #1,**

Thank you very much for your comments. I modified the manuscript as follows:

– RC: *Please comment and discuss on followings to clarify the implementation of the resolved gust approach in an incompressible solver:*

1. RC: *Loss-free transport of gust velocity through the flow field may require the free-stream velocity update not only on inflow boundary but on other far boundaries (top and sides) as well, or actually whole domain if necessary. How are the slip-condition velocities (or frestream conditions changing on these far boundaries? Gust model is changing inflow velocity i.e., freestream, and inflow velocity will effect all boundaries. Please clarify.* AC: The velocities on the slip-walls changes in the same way as the velocities in the whole domain. Due to the way, the implementation is performed in THETA, this happens without any user-interaction or additional user-implementation. A hint is given in section 4.1, page 7, line 18

2. RC: *And no-slip condition on the bottom boundary will result in somehow development of a Boundary Layer (BL) in the unsteady solution. Is there any development and effect of this BL on the tower and rotor, or flow field? Please clarify.* AC: Yes, I should have mentioned that the developing boundary layer has no effect on the HAWT performance because it remains below 1m in the entire flow domain. I will add a comment in section 5.1, page 9, line 25.

– RC: *Discussing clearly all the assumptions and difficulties encountered in such expensive unsteady computations will be helpful for future studies.* AC: I added the following sentences to the conclusion (section 6, page 18) "The assumptions, which are made in the paper can be summarized as

1. the wind speed is constant in height and time (except gust velocity)

2. the gust velocity is constant in height

3. the gust transport velocity is equal to speed of sound which is infinite

4. the boundary conditions of the flow domain are chosen to prevent the flow from escaping sideways.

The disadvantage of the resolved-gust approach clearly is the requirement of fine time steps and fine grids upstream of the geometry in question which increases the computational costs. The infinite speed of sound and its direct relation to the gust transport velocity is ambivalent. It leads to an inaccurate reproduction of the wind turbine wake transport on the one hand, while on the other hand, the response of the rotor loading to the gust velocity is immediately obtained. Clear advantages fo the resolved-gust approach are numerical stability and no artificial oscillations. Finally, the resolved-gust approach can be applied to any completed wind turbine U-RANS computation to obtain more insight of the wind turbine characteristics." Please also referre to the response to Referee #2, last point.

– Furthermore, I found some typos which were corrected.

– I deleted the sentence "The floor is defined as viscous wall." on page 6 as the information was given repeatedly in the paragraph.

**Dear Dear Anonymous Referee #2,**

thank you very much for your hints to improve the revised version of my paper. I updated the manuscript as follows:

- RC: *The paper has been improved since the original review, there are though a few points I would suggest to address before the final acceptance.* AC: Thank you

5 - RC: *Page 4, line 20: 'low dissipation' is mentioned twice.* AC: Please read carefulley. It is low dissipation low dispersion

- RC: *Grid resolution: The comment for the coarse grid still stands, the NREL Phase-VI investigation do not feature any grid dependency study. The fact that the results agree well with measured value do not guarantee grid independence. Additionally, the differencing scheme test in the referred paper is done at high AOA where cancelling errors might obscure the question. Comparing directly the Cp distributions, there are clear visual differences.* AC: Yes, a true grid
10 independence study has not been performed, yet. In the referenced paper, the high AOA has been chosen because it was the only test case that showed differences between the measurement and computation. In attached flow regions, even the result which were obtained with the first order upwind scheme matched the experiment perfectly (not shown in the referenced paper). Hence, the only possibility to show a difference between the schemes was in the partly separated flow region where a the difference between the second and third order schemes were small. I propse to add the performance
15 of a grid independency study of the test case to the to-do-list in the conclusion of the paper (section 9, page 19, line 2).

- RC: *Is the reference used for FAST correct, I would imagine that an older one exists?* AC: I updated the reference to an older version of 2013 even though I am using a the latest FAST version which includes the updates of 2015.

- RC: *Page 6: I do not think that a standard engineering viscous wall resolved condition will allow you to model atmospheric rough wall flow with roughness length 0.1 m and larger. Here typically Monin-Obukhov theory must be used.*
20 AC: I think that there is a small misunderstanding. Monin-Obukhov theory is used for non-neutral conditions and if the atmospheric boundary layer is actually resolved. I rather envisaged prescribing the neutral atmospheric boundary layer by logarithmic profiles than having a RANs solver resoving atmospheric flows. I reformulated the sentence on page 6 to clarify my point.

- RC: *Page 8, line 12: 't the computational time'. I would suggest to say: 't is the physical time simulated'* AC: Yes you
25 are right. Thank you for the hint.

- RC: *Page 12: There are basically no description on the turbine setup in the FAST model, is this based on the reference manual including airfoil data and more of Jonkman. Is the model including a Dynamic Inflow model, dynamic stall model . . . . So even though we see good comparisons, we do not know what we are comparing against.* AC: Of course it is as much of Jonkman and the reference manual as possible. Everything that is not is already mentioned in section 2.2.
30 I added the above information together with the parameter choices that you explicitly asked for to section 2.2.

- RC: *Why are there no thrust force shown for the FAST simulations in Figure 5?* AC: As the thrust does not add anything new to the discussion it is omitted to display it to keep a clear appearance of the figure.

- RC: *Page 14: It seems that the skin-friction ($C_f$) shown is basically the absolute value of the skin friction. I would suggest showing the Cf with a sign, easily indicating where separation is present. Additionally, the scaling of both the*
35 *Cp and Cf are not optimal for visual inspection.* AC: Yes, it is the absolute value of the skin-friction coefficient vector because in the present case no component of the 3D vector has had a dominant part that could be shown. Inversely, neglecting components of the $C_f$ vector would have led to missunderstandings in the results interpretation. The same holds for showing all components next to each other in both seperate or the same plots. It would have become a mess in presentation. Thus, in my oppinion the best way is to show the absolute friction force coefficient.
40 The differences between the scaling of figures 8 to 10 results from the very different magnitues in both $c_p$ and $c_f$. For all other issues about scaling, I would be glad if the reviewer would share his/her ideas more precisely.

– RC: *Page 18: Line 25: I would suggest removing the word 'perfect' and 'exactly'* AC: I changed perfect to 'very well' and exactly to 'very close'.

– RC: *Page 18: It would maybe also be appropriate to discuss whether the gain in details over the engineering modelling is worth the additional computational effort.* AC: Well, it is a three-stage process whereof the presented URANS computation is one of multiple contributions to the first stage. In the first stage, it is aimed for gaining more detailed knowledge about the behaviour of wind turbines during extreme gust events. This needs further computations of different modelling approaches and test conditions. In a second step, the engineering models can be adjusted to better predict the wind turbine behaviour during the design process. Consequently, the potential of weight reduction (and thus costs) and same reliability in wind turbine designs can be shown. I added this argument at the very end of my conlcusion (section 6, page 19). Please also refer to the response to Referee #1, last point.

– Furthermore, I found some typos which were corrected.

– I deleted the sentence "The floor is defined as viscous wall." on page 6 as the information was given repeatedly in the paragraph.

**Simulation of transient gusts on the NREL5 MW wind turbine using the U-RANS-solver THETA**

Annika Länger-Möller[1]

[1]DLR e.V.; Lilienthalplatz 7; 38108 Braunschweig

**Correspondence:** Annika Länger (annika.laenger@dlr.de)

**Abstract.** A procedure to propagate longitudinal transient gusts through a flow field by using the resolved-gust approach is implemented in the U-RANS solver THETA. Both, the gust strike of a $1 - \cos()$-gust and an extreme operating gust following the IEC 61400-1 standard are investigated on the generic NREL $5\,\mathrm{MW}$ wind turbine at rated operating conditions. The impact of both gusts on pressure distributions, rotor thrust, rotor torque, and flow states on the blade are examined and quantified. The flow states on the rotor blade before the gust strike, at maximum and minimum gust velocity are compared. An increased blade loading is detectable in the pressure coefficients and integrated blade loads. The friction force coefficients indicate the dynamic separation and re-attachment of the flow during the gust. Moreover, a validation of the method is performed by comparing the rotor torque during the extreme operating gust to results of FAST rotor code.

*Copyright statement.* The works published in this journal are distributed under the Creative Commons Attribution 3.0 License. This licence does not affect the Crown copyright work, which is re-usable under the Open Government Licence (OGL). The Creative Commons Attributions 3.0 License and the OGL are interoperable and do not conflict with, reduce or limit each other.

[revised manuscript text omitted]

---

## Author Response (AR3)

**Dear Mr. Soerensen,**

I corrected some minor grammatical issue and one citation in the introduction. At the bottom of page 8, the reference to equation 4 has been wrong and is now corrected to number 6. Other than that the paper remained unchanged. I thank you for your assistance with the review process. Sincerely A. Länger-Möller